# Kinetics of the LOV domain of ZEITLUPE determine its circadian function in *Arabidopsis*

**Ashutosh Pudasaini[1,2], Jae Sung Shim[3], Young Hun Song[3,4], Hua Shi[5], Takatoshi Kiba[6], David E Somers[5], Takato Imaizumi[3], Brian D Zoltowski[1,2]\***

[1]Department of Chemistry, Southern Methodist University, Dallas, United States; [2]Center for Drug Discovery, Design and Delivery, Southern Methodist University, Dallas, United States; [3]Department of Biology, University of Washington, Seattle, United States; [4]Department of Life Sciences, Ajou University, Suwon, Korea; [5]Department of Molecular Genetics, Ohio State University, Columbus, United States; [6]RIKEN Center for Sustainable Resource Science, Yokohama, Japan

**Abstract** A LOV (Light, Oxygen, or Voltage) domain containing blue-light photoreceptor ZEITLUPE (ZTL) directs circadian timing by degrading clock proteins in plants. Functions hinge upon allosteric differences coupled to the ZTL photocycle; however, structural and kinetic information was unavailable. Herein, we tune the ZTL photocycle over two orders of magnitude. These variants reveal that ZTL complexes with targets independent of light, but dictates enhanced protein degradation in the dark. In vivo experiments definitively show photocycle kinetics dictate the rate of clock component degradation, thereby impacting circadian period. Structural studies demonstrate that photocycle dependent activation of ZTL depends on an unusual dark-state conformation of ZTL. Crystal structures of ZTL LOV domain confirm delineation of structural and kinetic mechanisms and identify an evolutionarily selected allosteric hinge differentiating modes of PAS/LOV signal transduction. The combined biochemical, genetic and structural studies provide new mechanisms indicating how PAS/LOV proteins integrate environmental variables in complex networks.

**\*For correspondence:**
bzoltowski@smu.edu

**Competing interests:** The authors declare that no competing interests exist.

## Introduction

Organisms have developed signaling networks to measure and respond to daily (circadian) and seasonal (photoperiodic) alteration in environmental variables. Key to these circadian and photoperiodic responses is measurement of day length through complicated feedback loops involving sensory proteins. These sensory proteins involve members of the Period-ARNT-Singleminded and its Light, Oxygen, or Voltage (PAS/LOV respectively) subclass that couple photic-input to metabolic and stress pathways (*Lokhandwala et al., 2015*; *Sahar and Sassone-Corsi, 2009*; *Taylor and Zhulin, 1999*). Currently, how these signaling components are integrated is poorly understood due to a difficulty in decoupling photochemistry from allosteric protein changes and signal transduction. In plants and fungal species, LOV domain containing proteins act at signaling nodes to integrate sensory responses into circadian, reproductive and stress pathways (*Lokhandwala et al., 2015*; *Imaizumi et al., 2003*; *Somers et al., 2000*; *Zoltowski et al., 2007*). Central to their function are two key elements of the PAS family: (1) The ability to sense and respond to diverse environmental stimuli, and (2) The presence of multiple interaction surfaces that engage targets in a selective manner (*Figure 1*, *Figure 1—figure supplement 1*). The ability to trigger these elements with light has positioned LOV proteins as an allosteric model and enabled the development of LOV optogenetic

**eLife digest** Many living organisms track the 24-hour cycle of day and night via collections of proteins and other molecules that together act like an internal clock. These clocks, also known as circadian clocks, help these organisms to predict regular changes in their environment, like light and temperature, and adjust their activities according to the time of day.

Plants use circadian clocks to predict, for example, when dawn will occur and get ready to harness sunlight to fuel their growth. A plant called *Arabidopsis thaliana* has a light-sensitive protein called ZEITLUPE (or ZTL for short) that helps it keep its circadian clock in sync with the cycle of night and day. Previous studies have shown that light activates this protein causing part of it to change shape and then revert back after a period of about an hour and a half. However, it was unclear if this timing was important for ZEITLUPE to allow plants to keep track of time.

To help answer this question, Pudasaini et al. set out to identify a specific chemical event behind ZEITLUPE's changes in shape. A chemical bond forms when light activates ZEITLUPE, and it turns out that how long this bond lasts before it breaks plays an important role in allowing plants to maintain a 24-hour circadian clock. This chemical bond controls the shape changes that guide the protein's activities and, when Pudasaini et al. modified ZEITLUPE so that it took much longer for this bond to break, they could tune how fast the plant's internal clocks run. In essence, the time between the bond forming and breaking breaks acts like a countdown on a stopwatch, and it must be precisely timed to keep the clock in pace with the environment.

These findings improve our understanding of how light can regulate an internal biological clock. This improved understanding could, in the future, allow researchers to manipulate how plants and other organisms respond to their environment. This in turn could change how these organisms develop, and how much they grow. As such, extending these findings into agricultural crops may one day lead to new ways to increase crop yields.

tools (*Pudasaini et al., 2015*). However, limited understanding of how chemistry is linked to allostery and downstream signaling hampers our understanding of these systems.

Our current understanding of LOV signaling has benefited from detailed structural studies that revealed amino acid sites that tune LOV allostery without affecting LOV photocycle kinetics (*Zoltowski et al., 2007*; *Harper et al., 2004*; *Jones et al., 2007*; *Salomon et al., 2000*). A consensus mechanism is summarized. LOV domains are typified by blue-light induced formation of a C4a adduct between a conserved Cys residue and a bound flavin (FAD, FMN or riboflavin) cofactor (*Figure 1—figure supplement 1D*) (*Zoltowski and Gardner, 2011*). C4a adduct formation then drives rotation of a conserved Gln residue to initiate conformational changes within N- or C-terminal extensions (Ncap/Ccap) to the LOV core (*Zoltowski and Gardner, 2011*). These N/Ccap elements in turn regulate activity of effector domains or recruit proteins to Ncap, Ccap or $\beta$-sheet surfaces (*Figure 1—figure supplement 1A,C,E*). Upon incubation in the dark, or in the presence of UV-A light, the signaling adduct state spontaneously decays on a timescale of seconds to days (*Zoltowski and Gardner, 2011*; *Kennis et al., 2004*; *Pudasaini and Zoltowski, 2013*). In this manner, LOV proteins dictate transiently stable signaling states capable of switching between a distinct on and off state depending on lighting conditions (*Zoltowski et al., 2007*; *Halavaty and Moffat, 2007*; *Harper et al., 2003*; *Zoltowski et al., 2013*). These photoswitchable functions have made LOV proteins targets for optogenetic devices; however, currently we have limited understanding of the role of LOV photocycle kinetics for in vivo function. Genetic and photochemical studies of plant circadian and photoperiodic timing indicate *Arabidopsis* may function as a model organism for delineating the roles of photochemistry and allostery in LOV function.

In *Arabidopsis thaliana*, three LOV domain containing proteins ZTL, FLAVIN-BINDING, KELCH REPEAT, F-BOX 1 (FKF1), and LOV KELCH PROTEIN 2 (LKP2) act in circadian timing and seasonal flowering (*Baudry et al., 2010*). Among them, the genetic functions of ZTL and FKF1 are highly characterized. Recent research indicates that their divergent roles in the circadian clock and photoperiodic flowering may enable interrogation of how chemistry regulates PAS/LOV protein function to select for signaling pathways (*Figure 1—figure supplement 1B*) (*Zoltowski and Imaizumi, 2014*).

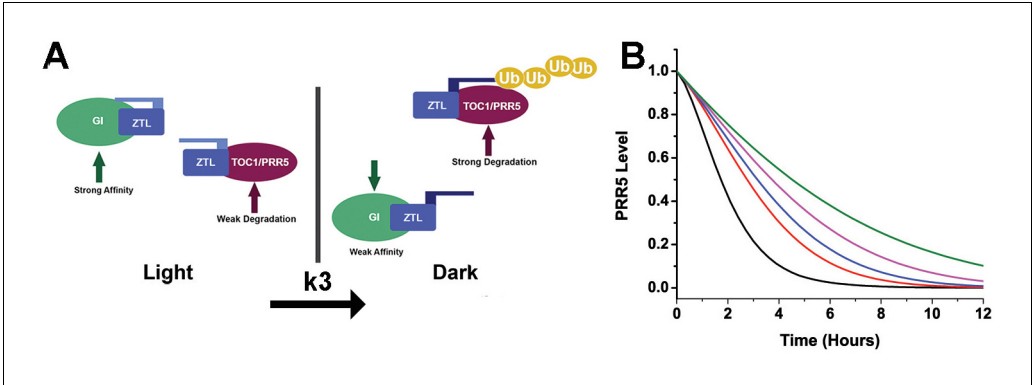

**Figure 1.** Models of ZTL photochemistry and regulation of circadian period. (**A**) In the light, ZTL associates with both GI and degradation targets (PRR5/TOC1). During the day, the strong affinity for GI allows GI, ZTL, TOC1 and PRR5 levels to rise. Upon dusk, the adduct form of ZTL decays with a rate constant $k_3$, leading to a conformational change. The conformation change decreases GI affinity and leads to ubiquitination of ZTL targets. (**B**) Modeling PRR5 degradation as a function of $k_3$ (see *Equation 1* and Materials and methods for model generation and parameter selection). Using $k_3$ for WT (black), G80R (red), V48I (blue), G46S:G80R (magenta) and V48I:G80R (green) leads to predictable changes in PRR5 degradation patterns. (*Figure 1—supplements 1*, *2* and *3*).

The following figure supplements are available for figure 1:

**Figure supplement 1.** LOV chemistry and signal transduction.

**Figure supplement 2.** Schematic diagram of the daily protein abundance profile of ZTL and its function in the circadian clock.

**Figure supplement 3.** Kinetics of ZTL Variants (A–D) Kinetics of ZTL adduct decay are determined from the absorbance recovery at 450 (black) and 478 nm (red).

These proteins retain analogous elements, where an N-terminal LOV domain regulates activity of a C-terminal E3 ubiquitin ligase to target clock proteins for degradation in a time dependent manner (*Baudry et al., 2010*; *Más et al., 2003a*). In addition, the LOV scaffold engages multiple proteins in a selective manner to regulate clock function (*Zoltowski and Imaizumi, 2014*). Despite conservation of domain elements, ZTL and FKF1 differ in their subcellular localization, degradation targets, and fundamental chemistry, thereby differentiating ZTL and FKF1 in circadian and photoperiodic timing (*Zoltowski and Imaizumi, 2014*).

Central to their function are key differences in photocycle kinetics. In FKF1, day-specific expression and a long-lived light-state species (τ = 62 hr) enable light-specific functions (*Imaizumi et al., 2003*; *Pudasaini et al., 2015*). In FKF1, photon absorption facilitates complex formation with GIGANTEA (GI) through its LOV domain (LOV-GI) to mediate degradation of CYCLING DOF FACTOR 1 (CDF1) and stabilization of CONSTANS (CO) (*Imaizumi et al., 2005*; *Sawa et al., 2007*; *Song et al., 2012*). Here, selection of degradation or stabilization appears to center on the domain involved in target recruitment, where the Kelch repeat domain (CDF1) specifies degradation and the LOV domain (CO) specifies stabilization (*Zoltowski and Imaizumi, 2014*; *Song et al., 2014*). Importantly, both functions require both light and GI.

In contrast, constitutive mRNA levels and a fast photocycle for ZTL (τ = 1.4 hr) suggest possible day (light) and evening (dark) functions (*Pudasaini and Zoltowski, 2013*). Light-state ZTL has two primary functions, formation of a LOV-GI complex to allow protein accumulation during the day via enhanced ZTL/GI stability (*Kim et al., 2013*, *2007*), and ZTL/GI-dependent destabilization of CO in the morning (*Song et al., 2014*). The latter results in antagonistic roles of ZTL and FKF1 in the photoperiodic response, and may contribute to a late-flowering phenotype in some ztl mutants (*Somers et al., 2004*). Throughout the night, ZTL mediates rapid degradation of the clock components, TIMING OF CAB EXPRESSION 1 (TOC1) and PSEUDO RESPONSE REGULATOR 5 (PRR5) (*Figure 1—figure supplement 2*) (*Kiba et al., 2007*; *Fujiwara et al., 2008*; *Más et al., 2003b*).

TOC1 protein levels contribute to the control of period length of the circadian clock (*Zoltowski and Imaizumi, 2014*). Impaired degradation of TOC1 by ZTL mutants lead to accumulation of TOC1 and PRR5 protein and a long-period phenotype consistent with strains harboring additional copies of TOC1 (*De Caluwé et al., 2016*). Interestingly, TOC1 degradation appears to be in competition with GI, occurs with an approximately 2 hr delay following dusk, and is enhanced in the dark (*Zoltowski and Imaizumi, 2014*; *Más et al., 2003a*; *Song et al., 2014*; *Kim et al., 2007*). Current models explain such behavior in TOC1/PRR5 turnover by imparting a differential function between day and night conditions (*De Caluwé et al., 2016*; *Pokhilko et al., 2013*), where TOC1/PRR5 are targeted for degradation by ZTL regardless of lighting conditions, but degradation activity is enhanced in the dark. How this is achieved at the molecular and chemical level is not well understood.

Herein, we present a comprehensive study of ZTL signaling to develop a broad understanding of blue-light regulation of circadian and photoperiodic timing and to understand the evolutionary basis for divergent functions of ZTL and FKF1. Therein, we tackle two outstanding questions in the PAS/ LOV field; (1) What purpose, if any, does the LOV photocycle lifetime play in biological function? Previous research proposed divergent roles for ZTL and FKF1 in the measurement of light intensity and day length (*Pudasaini and Zoltowski, 2013*), however no experimental validation of such a model has been available. Further, with the exception of a fungal system (*Dasgupta et al., 2015*), the role of LOV lifetimes in biology remains elusive. (2) How do PAS/LOV proteins signal to multiple interaction surfaces to allow signal integration? Based on structural studies of ZTL we have developed protein variants that decouple photocycle lifetimes from signal transduction. In this manner, we show a definitive role of LOV lifetime in circadian timing. Further, we provide an allosteric model of LOV signal transduction enabling selection of diverse protein:protein interactions.

## Results

Examination of mathematical models of circadian function reveals that ZTL photocycle kinetics may impart phase specific degradation of TOC1 and PRR5 (*De Caluwé et al., 2016*; *Pokhilko et al., 2013*). Namely by replacing indistinct day and night conditions by a difference in activity between light and dark-state ZTL, where the light-state inhibits ZTL activity in regards to degradation. In this manner, dark-state reversion activates ZTL during evening. Incorporating ZTL photochemistry into existing models of PRR5 results in *Equation 1* (see Mathematical model generation in methods), where $c_{PRR5}$ is the concentration of PRR5 protein, $k_1$ and $k_2$ the light and dark-state degradation constants and $k_3$ the rate of adduct decay for ZTL. An analogous equation, with equivalent $k_3$ dependence, can be derived for TOC1 (*Equation S12*), however, the expression pattern of TOC1 and complex regulation of TOC1 mRNA complicates analysis of TOC1 levels during the circadian cycle. For these reasons, we use the simpler PRR5 degradation data to mathematically test our model and use a qualitative analysis for examining effects of $k_3$ on TOC1 (see below).

$$\frac{dc_{PRR5}}{dt} = -\left\{(k_1 - k_2)e^{-k_3 t} + k_2\right\}c_{PRR5} \tag{1}$$

Examination of *Equation 1* provides distinct predictions on the effect of the ZTL adduct decay rate constant ($k_3$) on PRR5 and TOC1 levels. If rate-altering variants only affect photocycle kinetics, $k_3$ would dictate delays in PRR5/TOC1 degradation (*Figure 1*). Under these conditions a long photocycle would lead to increasing delays in PRR5 and TOC1 degradation leading to progressively longer circadian periods. Unfortunately, testing such predictions is complicated by difficulties in manipulating photochemical kinetics without altering allosteric regulation of ZTL function.

To delineate these aspects we focused on three residues in close proximity to the flavin active site that differ between ZTL and FKF1. These are G46 (Ser in FKF1) that lies adjacent to the active site Q154 implicated in Gln-flip signal transduction mechanisms (*Zoltowski and Gardner, 2011*), V48 (Ile in FKF1) that sterically interacts with C82 (*Zoltowski et al., 2009*), and G80 that lies in a GXNCRFLQ motif (X80=G for ZTL, X=R for FKF1) (*Figure 1—figure supplement 1E*). The V48 position is known to alter LOV photocycle kinetics by V/I/T substitutions (*Zoltowski et al., 2009*; *El-Arab et al., 2015*; *Christie et al., 2007*; *Lokhandwala et al., 2016*). This site has been exploited in optogenetic tools to tune signal duration under the presumption that it does not affect allosteric regulation of protein function (*Strickland et al., 2012*). In contrast, to the best of our knowledge,

the G80 position has not been exploited for tuning LOV kinetics. This likely is due to *Arabidopsis* ZTL being unusual amongst LOV proteins in containing a Gly residue at this position. As a consequence of G80, ZTL expresses poorly in *E. coli* and is largely confined to inclusion bodies (see Materials and methods). Since, G80 is unique to *Arabidopsis* ZTL, and R/Q/K substitutions are permitted in other ZEITLUPE and LOV proteins (see Discussion), it is unlikely to affect allostery and may function as a site to uniquely affect photocycle kinetics.

Consistent with our predictions, kinetics of V48I and G80R differentiate ZTL and FKF1 type chemistry. Specifically, variants extend the ZTL lifetime from 1.4 hr (WT) to 6.6 (G80R) or 10.7 (V48I) hours (*Figure 1—figure supplement 3* and *Table 1*). Double variants G46S:G80R and V48I:G80R function cooperatively to extend the ZTL photocycle to 21 hr or >65 hr respectively, very similar to WT FKF1 (62.5 hr) (*Pudasaini and Zoltowski, 2013*) (*Figure 1—figure supplement 3* and *Table 1*). In addition, the difference in values for $k_3$ in these variants are sufficient to elicit theoretical differences in PRR5/TOC1 degradation rates (*Figure 1B*). To verify that rate-altering variants do not also perturb allosteric regulation of ZTL function it is essential to obtain crystal structures of ZTL variants and to map an allosteric trajectory that couples C4a adduct formation to regulation of ZTL function. Towards these aims we solved 2.5 Å, 2.6 Å, and 2.1 Å crystal structures of dark-state WT ZTL, G80R and V48I:G80R (residues 29–165) respectively (see *Table 2*). In addition, the long-lived V48I:G80R variant enabled direct crystallization (2.3 Å) of the light-state adduct allowing for interrogation of signal transduction pathways.

## ZTL structures reveal an unusual mechanism of signal transduction in LOV proteins

All ZTL variants form solution dimers, crystallize in the same space group (P3121) and demonstrate topologically equivalent structures consistent with PAS/LOV proteins (*Figure 2*, *Figure 2—figure supplements 1* and *2*). These structures reveal functional differences from all known LOV structures that differentiate the effects of V48I and G80R on ZTL signaling mechanisms. Namely, G80R structures are analogous to WT in all manners. The primary difference is the formation of a π-cation interaction that stabilizes the Dα/Eα linkage and C82 that is necessary for C4a adduct formation (*Figure 2B*). Increased rigidity of C82 imposed by the π-cation interaction is consistent with having an effect only on photocycle kinetics. In contrast, examination of V48I containing structures reveals distinct differences that identify allosteric signal transduction mechanisms to the N/Ccap. These mechanisms identify V48I as a residue that disrupts allosteric regulation of ZTL function. Below, we provide detailed analysis of ZTL in dark and light states to highlight these functional differences. We focus on two reported aspects of ZTL group function, LOV:LOV mediated homodimerization and allosteric regulation of Ncap and Ccap elements.

## LOV dimerization

Previous solution studies of ZTL group proteins suggest they function as obligate dimers (*Ito et al., 2012*; *Nakasako et al., 2005*), however ZTL group dimers have not been observed in vivo and their function is unknown (*Han et al., 2004*). Consistent with solution studies, the crystallographic lattice is defined by two sets of anti-parallel dimers formed by extensive contacts along the β-scaffold. We term these interfaces the 'compact' and 'elongated' dimers based on a 2 Å translation that is coupled to the degree of order present within the Ncap (*Figure 2*). In the compact dimer, clear density

**Table 1.** Kinetics of thermal reversion for LOV constructs and variants at 296 K. Uncertainty is depicted as the standard deviation from three replicates.

| Construct | Time Constant, 1/k3 (hrs) |
| --- | --- |
| WT ZTL 29–165 | 1.4 ± 0.1 |
| G80R | 6.6 ± 0.1 |
| V48I | 10.7 ± 0.8 |
| G46S:G80R | 21 ± 3 |
| V48I:G80R | 65 hr<τ |

**Table 2.** Data collection and refinement statistics (molecular replacement).

| | WT ZTL Dark | G80R Dark | ZTL-Dark V48I:G80R | ZTL-Light V48I:G80R |
|---|---|---|---|---|
| PDB ID | 5SVG | 5SVU | 5SVV | 5SVW |
| Data collection | | | | |
| Space group | P3(1)21 | P3(1)21 | P3(1)21 | P3(1)21 |
| Cell dimensions | | | | |
| $a$, $b$, $c$ (Å) | 85.0, 85.0, 199.5 | 85.4, 85.4, 200.0 | 85.4, 85.4, 198.8 | 86.2, 86.2, 200.5 |
| abg (°) | 90, 90, 120 | 90, 90, 120 | 90, 90, 120 | 90, 90, 120 |
| Resolution (Å) * | 2.5 (2.59–2.5) | 2.6 (2.69–2.6) | 2.10 (2.18–2.1) | 2.29 (2.35–2.29) |
| $R_{sym}$ or $R_{merge}$ | 7.6 (27.9) | 12.8 (23.4) | 5.6 (25.1) | 6.1 (19.9) |
| $I$ / $sI$ | 35.0 (11.1) | 55.4 (22.7) | 36.6 (12.6) | 19.8 (5.5) |
| Completeness (%) | 97.1 (98.6) | 97.9 (99.8) | 99.3 (99.0) | 87.0 (84.3) |
| Redundancy | 9.0 | 10.0 | 7.9 | 2.6 |
| Refinement | | | | |
| Resolution (Å) | 2.5 | 2.6 | 2.1 | 2.3 |
| No. reflections | 28790 | 26153 | 49510 | 34582 |
| $R_{work}$ / $R_{free}$ | 17.8/24.0 | 16.4/23.2 | 16.3/20.0 | 16.2/22.8 |
| No. atoms | | | | |
| Protein | 3964 | 3990 | 3986 | 3950 |
| Ligand/ion | 124 | 124 | 148 | 124 |
| Water | 142 | 139 | 343 | 269 |
| $B$-factors | | | | |
| Protein | 47.3 | 46.8 | 34.6 | 37.5 |
| Ligand/ion | 35.2 | 35.4 | 27.6 | 27.5 |
| Water | 46.8 | 42.4 | 41.6 | 39.6 |
| R.m.s. deviations | | | | |
| Bond lengths (Å) | 0.014 | 0.014 | 0.012 | 0.014 |
| Bond angles (°) | 1.58 | 1.61 | 1.46 | 1.65 |
| Ramachandran outliers | 2 (0.4%) | 3 (0.62%) | 1 (0.21%) | 1 (0.21%) |

*Highest-resolution shell is shown in parentheses.

for Ncap residues (31-43) is observed in one molecule that contacts the helical interface of the adjacent monomer (*Figure 2C,D*). In contrast, the elongated dimer contains no density for Ncap residues (*Figure 2E,F*). A stabilizing element in both dimers is a hydrophobic core composed of a tetrad of Ile residues (I151 and I153).

Additional contacts along the helical interface define a possible secondary dimer. The helical dimer is parallel in orientation and involves contacts between the E-F helices and associated loop. Specifically, R95 forms a salt bridge with the phosphate moiety of FMN in the neighboring molecule (*Figure 2—figure supplement 2*). However, based on two lines of evidence, we conclude that the β-scaffold interface represents the solution ZTL and FKF1 homodimers. First, FKF1 lacking the entire E-F loop and ZTL variants that disrupt R95 contacts remain dimeric (*Figure 2—figure supplement 2*) (*Nakasako et al., 2005*). Second, I151R (and I160R in FKF1) variants abolish dimer formation in vitro (*Figure 2A* and *Figure 2—figure supplement 1*). Thus, ZTL and FKF1 solution dimers are formed by equivalent anti-parallel contacts along the β-scaffold, similar to other PAS/LOV proteins (*Card et al., 2005*).

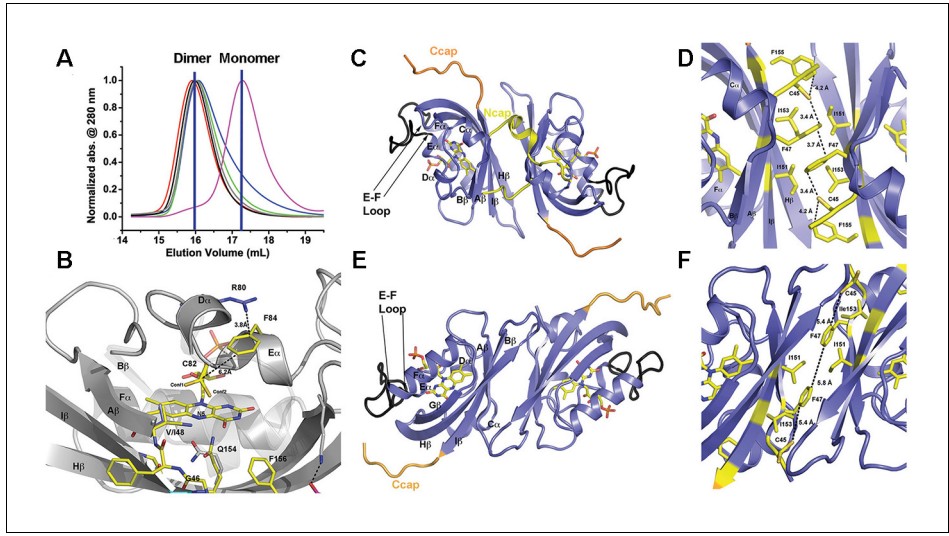

**Figure 2.** Structural analysis and LOV dimer formation in ZTL. (**A**) G80R (dark-state, black; light-state (grey), WT (red), V48I:G80R (green), G46S:G80R (blue) all elute as dimers with apparent MWs of 38–41 kDa compared to the expected monomer of 16 kDa. Multi-Angle-Light-Scattering (MALS) confirms dimer formation in WT 29–165 (absolute MW 33 ± 2 kDa) and 16–165 constructs (See *Figure 2—figure supplement 1*). Introduction of an I151R abolishes dimer formation (magenta; apparent MW = 22 kDa). (**B**) Structure of ZTL active site (yellow) and residues involved in structural or kinetic modulation of signaling (purple). R80 within Dα forms a π-cation interaction with F84 directly above the photoreactive C82, resulting in steric stabilization of adduct formation. The observed steric stabilization through the π-cation interaction is consistent with the longer photocycle in G80R. V48I positions the additional methyl group into a pocket adjacent to N5, C82 and Q154. Comparisons of AsLOV2 structures (white; buried conformation), dark-state ZTL (yellow; exposed conformation) demonstrates that V48I can impact the position of Q154 between buried and exposed conformations. Movement of Q154 correlates with movement of F156 in Iβ. (**C–F**) ZTL monomers are defined by an antiparallel β-sheet flanked by a series of α-helices (Cα, Dα, Eα, Fα). The helices cradle the photoreactive FMN adjacent to C82 (Eα helix). ZTL contains a 9-residue insert linking the E-F helices that accommodates the adenine ring of FAD in some LOV proteins (black) (*Zoltowski et al., 2007*). N- and C-terminal extensions (Ncap/Ccap; yellow) are largely disordered; however, a short helix within the Ncap reaches across a dimer interface in some molecules to form contacts between the Cα and Dα helices. Two dimer interfaces are formed through the β-scaffold in ZTL. The compact dimer (**C,D**) differs from the elongated dimer (**E,F**) by a 2.0 Å translation along the β-sheet. Key residues in the dimer interface are shown in yellow. The translation disrupts a network of sulfur-π and π-π interactions involving C45 and F47, centered around I151. (*Figure 2—figure supplements 1* and *2*).

The following figure supplements are available for figure 2:

**Figure supplement 1.** Dimerization of ZTL/FKF1/LKP2 LOV domains.

**Figure supplement 2.** Dark-state structure of ZTL 29–165 and helical dimer.

## N-terminal CGF motif defines a locus of signal transduction that differentiates ZTL from known LOV signaling mechanisms

Given the unknown role of ZTL dimers in vivo and the known role of Ncap and Ccap elements in LOV allostery, we turned our attention to the structural differences between the different dimers. Close examination of residues linking the Ncap and Ccap to the active site FMN identifies distinguishing interactions that may be involved in signal transduction. The loss of Ncap density in the elongated dimer directly follows a CGF motif (C45-G46-F47-(V48)) that links the Ncap, FMN binding pocket and the V48 position (*Figure 3*). An analogous hinge involving a Cys residue is known to mediate signal transduction in fungal LOV proteins, where the hinge directs both a conformational change and integration of oxidative and osmotic stress (*Lokhandwala et al., 2015*; *Zoltowski et al., 2007*, *2009*; *Lokhandwala et al., 2016*; *Lamb et al., 2009*; *Zoltowski and Crane, 2008*). Further, the CGF motif differentiates ZTL and FKF1, where FKF1 contains the G46S and V48I (equivalent)

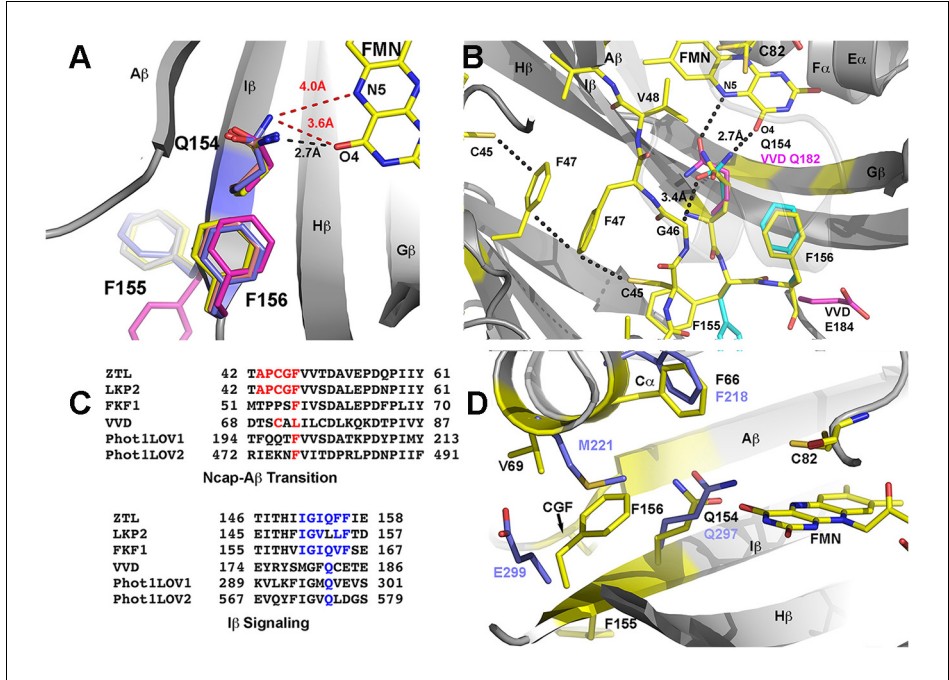

**Figure 3.** Q154 links Ncap, Ccap and helical elements. (**A**) Q154 exists in multiple conformations in WT ZTL structures. They differ in interactions with the active site flavin. An exposed conformation forms strong H-bonds to the O4 position (black dotted line). A buried conformation forms weaker interactions (3.6 Å) with O4 that leads to closer interactions at N5 (4.0 Å; red dotted line). The altered conformation is coupled to movement of F156 into the active site and multiple conformations of F155, forming a QFF motif. The altered conformations define ZTL signaling as distinctly different than existing LOV structures. (**B**) The unusual orientations of Q154 differ from other LOV proteins that typically show strong interactions near N5 (VVD; magenta). The heterogeneous conformations of Q154 directly abut G46 in a CGF motif allowing formation of the sulfur-π and π-π interactions involving C45 and F47. F156 then adopts a buried conformation in contrast to the equivalent residue in VVD (E184) (**C**) Sequence alignments of LOV proteins depict conserved elements within the CGF motif (red) and QFF motif (blue) in *Arabidopsis thaliana* ZTL, LKP2, FKF1, phototropin 1 LOV1 and LOV2. Sequence conservation indicates divergent signaling mechanisms within the ZTL/FKF1 family compared to existing LOV allostery models. (**D**) Comparisons of ZTL (yellow, black lettering) and *Arabidopsis thaliana* phototropin 1 LOV1 (PDB: 2Z6C; blue). The altered conformation of Q154 draws F156 into the active site. The buried conformation of F156, leads to movement of Cα (F66, V69). (*Figure 3—figure supplement 1*)..

The following figure supplement is available for figure 3:

**Figure supplement 1.** Heterogeneous orientations of Q154.

---

mutations, thereby highlighting the region as a possible factor regulating the divergent functions of these closely related proteins.

In ZTL, signal transduction diverges from known LOV signaling mechanisms (*Figures 3* and *4*). In contrast to other LOV proteins, where a conserved Gln (Q154) acts as a molecular bridge linking the FMN to a residue in Aβ, the dark-state structure of ZTL contains a heterogeneous orientation of the active site Q154 (*Figure 3* and *Figure 3—figure supplement 1*) (*Zoltowski et al., 2007*; *Halavaty and Moffat, 2007*; *Möglich and Moffat, 2007*). The altered orientation is coupled to contacts in the Ncap, Ccap and helical interface. Given that all three sites have been implicated in PAS/LOV signaling (*Harper et al., 2004*; *Card et al., 2005*; *Zoltowski and Crane, 2008*; *Partch et al., 2009*), we examine each interaction below. We define them as the Aβ-Ncap hinge, Iβ-Ccap hinge and Cα (helical interface) to specify structural elements that may be involved in signal transduction.

In ZTL, Q154 does not adopt a specific orientation as observed in all other LOV structures, rather it varies in all four molecules in the asymmetric unit of both WT-ZTL and G80R (*Figure 3A* and *Figure 3—figure supplement 1*). The altered conformations contact the Aβ-Ncap hinge through G46 in

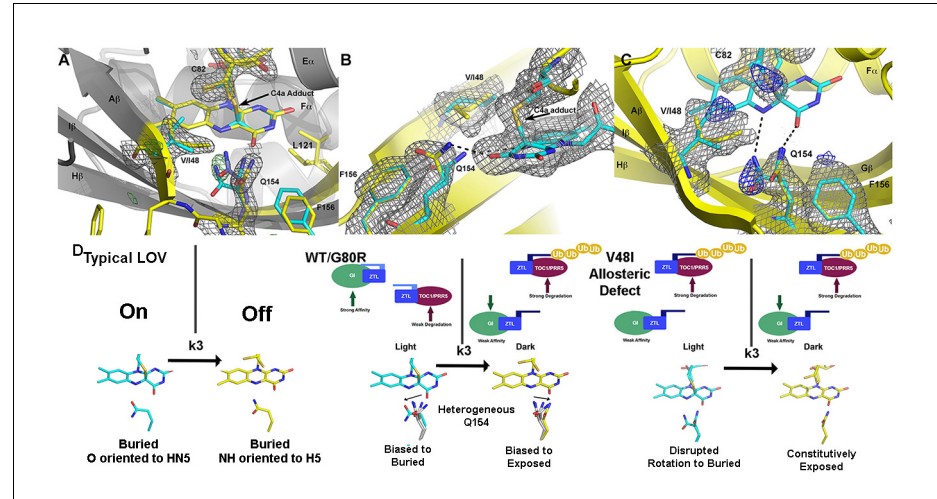

**Figure 4.** Structural effects on ZTL chemistry and signaling. (**A**) Comparison of dark-state V48I:G80R (yellow), light-state V48I:G80R (cyan) and dark-state AtLOV1 phototropin 1 (purple: PDB ID: 2Z6C) molecules. 2Fo-Fc (2.0 σ grey mesh) and Fo-Fc (3.0 σ green mesh) are depicted for dark-state V48I:G80R. Lack of density for an adduct is consistent with minimal population of the light-state species. Density shows clear selection of the I48 methyl group towards Q154. Some residual density is present in the buried conformation of Q154 that indicates either partial occupancy of the site in the dark, or residual light-state conformations. The buried conformation correlates with the orientation of Q154 in all other LOV structures as depicted by AsLOV2. Electron density for FMN is excluded to allow clear observation of electron density for active site side-chains. (**B**) Light state crystal structure of V48I:G80R, 2Fo-Fc data shown at 2.0 σ show (grey mesh) clear electron density for C4a adduct formation. (**C**) Rotated view of the active site of the light-state ZTL molecule. Modeling of Q154 in the exposed conformation (yellow; panel C) results in Fo-Fc (blue mesh; panel C) density at 3.0 σ for the light-state molecule. Electron density for the FMN is excluded for clarity. The data confirms rotation of Q154 to a buried conformation (cyan) following adduct formation. Rotation of Q154 is coupled to rearrangement of V/I48. In V48I, the additional methyl groups blocks rotation, partially inhibiting population of the buried conformation of Q154 (cyan conformation). (**D**) Predicted divergent ZTL model of allostery and signal transduction based on the integrated structural, mutational and in vivo data. Orientations of WT/G80R Q154 are derived from the dark-state G80R ZTL structure with buried and exposed conformations shown for reference from V48I:G80R. WT/G80R and V48I:G80R deviate from typical LOV models (derived from VVD: light PDBID 3RH8, dark PDBID 2PD7) on the position of Q154. WT/G80R ZTL retains a heterogeneous orientation of Q154. We propose that Q154 is heterogeneous regardless of lighting conditions, but biased towards the buried conformation in the light. At dusk adduct decay, with rate constant $k_3$, causes the Q154 conformation to be biased towards an exposed conformation, accelerating ubquitination of protein targets. For V48I, I48 selects the exposed conformation in the dark and leads to only partial burying of Q154 (shown in C, (D), leading to constitutively high ubquitination activity that mimics the dark-state of WT-ZTL. (*Figure 4—figure supplements 1 and 2*).

The following figure supplements are available for figure 4:

**Figure supplement 1.** Effect of ZTL variants on protein complex formation.

**Figure supplement 2.** Predicted differences in G46 mutants.

---

the CGF motif (Ncap; *Figure 3B*). Here, Q154 abuts (3.4 Å) the Cα carbon of G46 (Ser in FKF1). The lack of a side chain at G46 permits favorable interactions for Q154 at the flavin O4 position and allows insertion of F156 into the flavin active site. In this manner an unusual orientation of Q154 links the Iβ-Ccap hinge and helical interface through a QFF motif (Q154-F155-F156). In most LOV proteins, the equivalent residue to F156 is hydrophilic and adopts a solvent exposed position (*Figure 3*). The resulting cavity occupied by F156 is filled by a Met or Leu residue (V69 in ZTL) from Cα (*Figure 3B,D*). In ZTL, the altered orientation of Q154 clashes with typical orientations of Cα. Steric clashes lead to movement of conserved F66 and rotation of V69 away from Q154 and the FMN binding pocket.

The combined interactions stemming from Q154 positions it ideally to have concerted conformational changes within Ncap and Ccap elements. Although, the Gln locus has been cited as the primary source of signaling in LOV proteins (*Zoltowski et al., 2007*; *Halavaty and Moffat, 2007*; *Freddolino et al., 2013*), the lack of interactions near N5 and A$\beta$ is atypical. Thus, it is likely that ZTL does not involve an H-bonding switch in signal transduction and that canonical mechanisms of LOV signaling may not be universally conserved. Rather, we propose that steric interactions involving the flavin O4 position and G46 shift a dynamic equilibrium in the Q154 position to stabilize the ZTL protein in a light-state configuration. We posit that the heterogeneous orientation of Q154 destabilizes ZTL in the absence of covalently attached FMN, consistent with in vitro studies demonstrating a tendency to lose flavin rapidly. Upon light activation, ZTL is stabilized by covalent attachment of FMN (through C82) and results in movement of Q154 that necessitates rearrangement of the CGF, QFF and C$\alpha$ sites to elicit signal transduction.

Dark and light state structures of V48I:G80R confirm concerted movement of Q154 that is linked to conformational changes in V/I48. The Q154 conformation is not heterogeneous in V48I:G80R, rather, the Ile side chain sterically directs Q154 to an exposed conformation near O4 that has not been observed in other LOV proteins. Comparing the dark-state to V48I:G80R grown in the light, confirms density for the C4A adduct and confirms direct crystallization of the light-state (*Figure 4*). Difference density maps of the light-state molecule indicate that adduct formation is coupled to rotation of Q154 to a buried position (*Figure 4C*). Unexpectedly, rotation of Q154 requires concerted movement of I48 that reorients the Ncap. Thus, the presence of G46 selects for a heterogeneous conformation of Q154 and V48I biases the heterogeneous Q154 towards the exposed conformation in the dark. In the light, V48I disrupts rotation of Q154 to the buried conformation, thereby leading to a predominantly exposed conformation regardless of lighting conditions. Thus, unexpectedly, V48I is both an allosteric variant and alters photocycle kinetics. In particular, V48I retains an ability to form the covalent C4a adduct, which is more stable than WT (V48I $k_3$ = 0.094 hr$^{-1}$; WT $k_3$ = 0.71 hr$^{-1}$). However, in V48I allosteric regulation of the Ncap through Q154 is disrupted leading to V48I selecting for a distinct exposed Q154 conformation despite adduct formation (*Figure 4*).

Based on our structural results we can refine our models on how ZTL rate altering variants will perturb ZTL function. G80R does not impact ZTL structure or allosteric regulation of Ncap or Ccap elements. In this manner, G80R acts as a photocycle variant only and allows direct testing of LOV photocycle kinetics on targeted degradation of PRR5 and TOC1 (see *Equation 1*). In contrast, despite being photochemically active and stabilizing the C4a adduct, V48I blocks allosteric regulation of the Ncap through incomplete rotation of Q154 and selection of a distinct exposed conformation of Q154. Thus, we propose that V48I is an allosteric variant that mimics the degradation-active dark state. Combining our proposed mechanism with *Equation 1* above and existing literature on light-dark formation of ZTL:GI complexes we make the following testable hypotheses: (1) V48I should demonstrate weaker interactions with GI. (2) V48I should show constitutive activity regardless of lighting conditions in targeting PRR5 and TOC1 for degradation. Thus, PRR5 and TOC1 levels should be constitutively low in variants containing the V48I mutation.

## V48I disrupts ZTL:GI interactions

We examined ZTL variants for their ability to form light and dark regulated complexes with GI and ARABIDOPSIS SKP1-LIKE 2 (ASK2) of the SCF complex. CoIP results confirm that G80R retains light regulated complex formation with GI comparable to WT. In contrast, Ncap variants G46S and V48I both alter light driven complex formation with GI (*Figure 4—figure supplement 1*). Whereas, V48I leads to dampened light-driven formation of the ZTL:GI complex, G46S enhances GI complex formation in both the dark- and light-states (*Figure 4—figure supplement 1A*). These results support our allosteric model of ZTL regulation and demonstrate that Ncap variants decouple allosteric regulation of signal transduction from photochemical formation of the C4a adduct. Further, the data implicates light-driven conformational changes near the Ncap in dictating GI interactions. Specifically, where V48I mimics the weak GI interacting dark-state and G46S mimics the strong GI interacting light state. We note that the G46S results reported here diverge from the effects of the G46E mutation reported by *Kim et al. (2007)*, where G46E abolishes GI interactions due to apparent misfolding of the LOV domain. These results are informative on the nature of mutations at the G46 site, namely the long side chain present in a G46E variant leads to steric clashes and likely would force E46 into a

hydrophobic pocket (see *Figure 4—figure supplement 2*). We contend such clashes leads to the destabilization of the LOV domain and abolition of GI interactions as reported previously for G46E (*Kim et al., 2007*). In contrast, the shorter sidechain in G46S, can be easily accommodated by a subtle rotation of the active site Q154 towards the proposed light-state buried conformation (modeled in *Figure 4—figure supplement 2*). Such results are consistent with enhanced GI interactions in G46S. Combined, the results implicate the G46/V48 locus in ZTL allostery and regulation.

In contrast to GI, where G46/V48 mutations affect function in a light/dark manner, all ZTL variants complex with ASK2 in a light-independent manner comparable to WT (*Figure 4—figure supplement 1B*). The protein:protein interaction data confirms G80R behaves as WT in known biochemical functions of ZTL and only differs in photocycle kinetics. In contrast, V48I acts as an allosteric variant mimicking the dark-state conformation. Based on these results and previously published in vivo studies showing enhanced degradation activity in the dark (*Más et al., 2003a*), we have tools to test the effect of photocycle kinetics (G80R) and Ncap allostery (G46S/V48I). In this manner, G80R variants isolate effects of photocycle kinetics allowing testing of predictions based on *equation 1*.

The decrease in GI affinity in the V48I variant could complicate in vivo phenotypes for these variants. Literature indicates ZTL stability is dictated by GI interactions (*Kim et al., 2007*). Thus, we could expect low ZTL levels in V48I variants. These low ZTL levels would act in opposition to any increased targeted degradation of PRR5/TOC1 and could mask allosteric phenotypes. Based on our experimental conditions, we do not expect any complications to result. Prior studies indicate that ZTL and GI reciprocally stabilize each other (*Kim et al., 2013*, *2007*), but increased ZTL stability occurs in a circadian phase dependent manner (*Kim et al., 2003*). Light does enhance ZTL:GI affinity and reciprocal stability by three fold, but ZTL retains some stabilization during the subjective circadian day regardless of lighting conditions (*Kim et al., 2007*, *2003*). These previous findings suggest that in consideration of the ZTL:GI equilibrium, GI is limiting except during the subjective circadian day. During the day, GI levels rise sufficiently to shift the equilibrium to saturate the ZTL:GI complex regardless of lighting conditions. Given that our V48I variant retains light-state affinity comparable to WT dark-state protein, GI expression during the subjective day under LL conditions should rescue the decrease in affinity. Concomitant with ZTL-ox, reciprocal stabilization of ZTL/GI should lead to high ZTL levels and ZTL should no longer oscillate. Indeed, under our experimental conditions cycling of ZTL protein is lost and V48I variants show enhanced stability in vivo (*Figure 5—figure supplement 1*), thereby the role of GI in ZTL stability is masked under our conditions and our results likely reflect the effect of ZTL photochemistry (G80R) and allosteric activation of PRR5/TOC1 degradation (V48I) independent of the effects on GI binding.

Based on the data above we make the following predictions. In comparing G80R-ox to WT-ox under LD conditions, the fast decay of WT-ox will lead to higher populations of the active-dark state early in the evening. In contrast, slower adduct decay in G80R will lead to prolonged population of the less active light-state and delays in ZTL degradation activity. As a result, we should observe an enhanced delay in PRR5/TOC1 degradation in G80R variants as shown in *Figure 1* and *Figure 5—figure supplement 2*. For V48I and V48I:G80R, the allosteric effects should enhance degradation of PRR5 and TOC1 leading to constitutively low PRR5/TOC1 levels.

## ZTL variants alter targeted degradation of clock components

To test our predictions, we constructed *Arabidopsis* transgenic ZTL overexpression lines (ZTL-ox) containing WT, V48I, G80R and V48I:G80R. G46S variants were excluded due to poor yields from *E. coli* that render solution or structural information intractable for the isolated G46S variant (see Materials and methods). These constructs were then examined for their effect on PRR5 and TOC1 degradation as well as circadian period and amplitude. All transgenic lines were selected based on similar *ZTL* transcript levels and ZTL protein levels were measured (*Figure 5—figure supplement 1*).

Consistent with our predictions, G80R-ox (#22) leads to delayed degradation of PRR5 and TOC1 under LL and LD (*Figure 5*). Specifically, despite being overexpressed (*Figure 5—figure supplement 1A*), G80R-ox variants demonstrate peak amplitudes of TOC1 consistent with WT (*Figure 5A, B*). Further, comparison of apparent rate constants for PRR5 degradation confirms a direct effect of ZTL photocycle kinetics on PRR5 degradation, where G80R #22 (0.34 hr$^{-1}$ LD, 0.14 hr$^{-1}$ LL) exhibits smaller rate constants compared to WT-ox (0.5 hr$^{-1}$ LD, 0.13 hr$^{-1}$ LL) under LD conditions where $k_3$ is most relevant (*Table 3*). These results are consistent with a more active dark-state ZTL. Thus, ZTL photocycle kinetics regulate degradation of TOC1/PRR5.

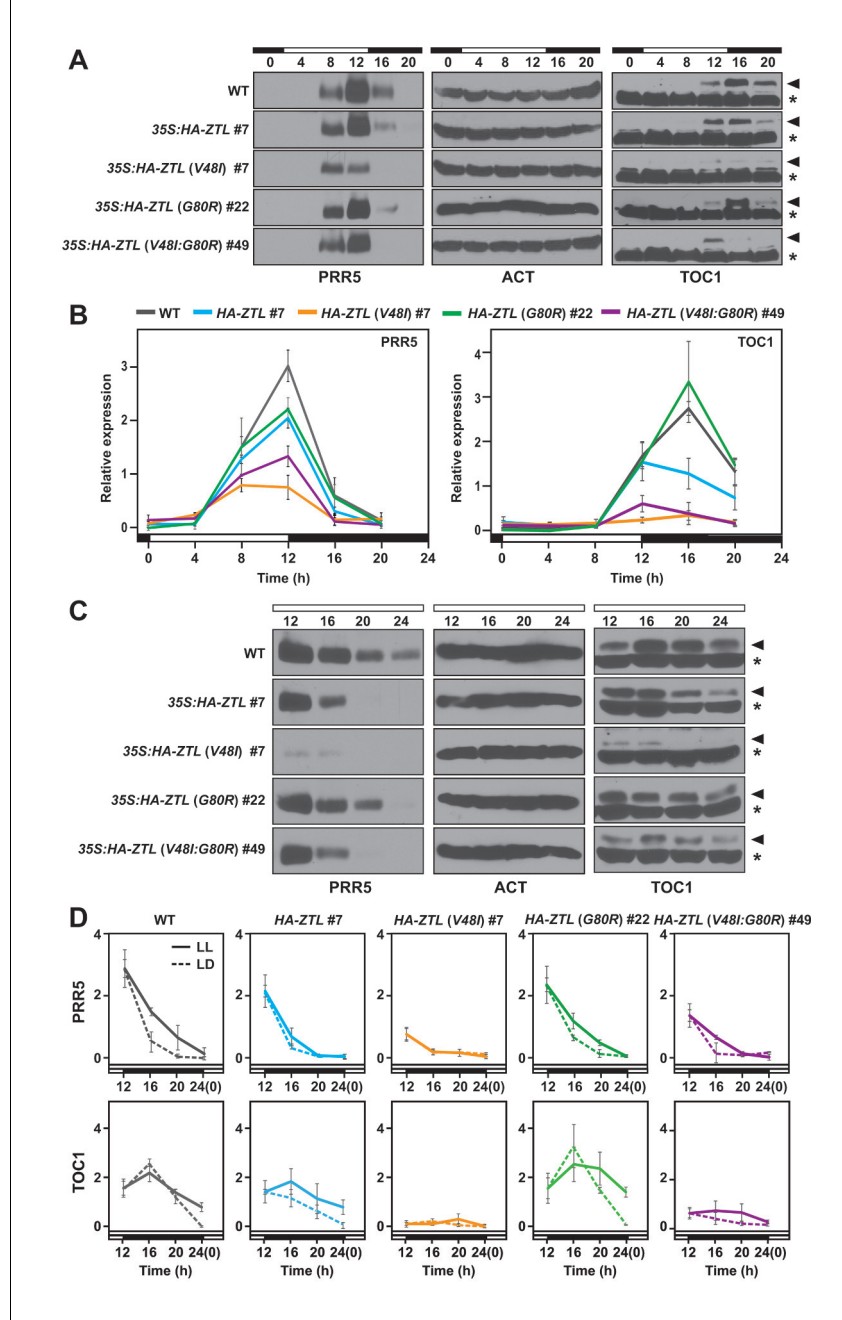

**Figure 5.** Diurnal and circadian expression profiles of PRR5 and TOC1 proteins in *ZTL* variant overexpressors. (**A**) PRR5 and TOC1 protein levels were analyzed in WT, *35S: HA-ZTL*, *35S: HA-ZTL (V48I)*, *35S: HA-ZTL (G80R)* and *35S: HA-ZTL (V48I:G80R)* under 12L/12D conditions. Actin (ACT) was used as a loading control for PRR5. Arrowhead indicates the band corresponding to TOC1 protein, while an asterisk indicates a nonspecific cross-reacting band, which is used as a loading control. (**B**) Relative expression level of PRR5 and TOC1 were determined in WT, *35S: HA-ZTL*, *35S: HA-ZTL (V48I)*, *35S: HA-ZTL (G80R)* and *35S: HA-ZTL (V48I:G80R)* under 12L/12D conditions. Actin and the TOC1 nonspecific bands were used for normalizing protein loadings for quantification of PRR5 and TOC1. The data represent the averages ±SEM obtained from three biological replicates. (**C**) PRR5 and TOC1 protein levels were analyzed in WT, *35S: HA-ZTL*, *35S: HA-ZTL (V48I)*, *35S: HA-ZTL (G80R)* and *35S: HA-ZTL (V48I:G80R)* during the subjective night under constant light conditions. (**D**) Relative levels of PRR5 and TOC1 proteins were determined in WT, *35S: HA-ZTL*, *35S: HA-ZTL (V48I)*, *35S: HA-ZTL (G80R)* and *35S: HA-ZTL (V48I:G80R)*. Dashed lines represent protein levels under 12L/12D conditions, while solid lines

*Figure 5 continued on next page*

*Figure 5 continued*
represent protein levels under constant light conditions. The data represent the averages ±SEM obtained from three biological replicates. (*Figure —figure supplement 1* and *2*).

The following figure supplements are available for figure 5:

**Figure supplement 1.** Relative expression levels of clock proteins.

**Figure supplement 2.** Comparison of model to PRR5 degradation in vivo.

Examining V48I and V48I:G80R variants confirms a role of V48I in altering light-dark regulation of ZTL activity. Strains harboring V48I and V48I:G80R show constitutively low levels of PRR5 and TOC1, consistent with high degradation activity regardless of lighting conditions (*Figure 5*). Both variants show degradation rate constants for PRR5 of ~0.8–1 hr$^{-1}$ under LD conditions, similar to maximum rate constants (0.8 hr$^{-1}$) predicted from computation models of clock function (*De Caluwé et al., 2016*; *Pokhilko et al., 2013*) (see Materials and methods and *Table 3*). V48I and V48I:G80R also exhibit high activity under LL conditions, demonstrating a partial loss of light-dark regulation (*Figure 5* and *Table 3*). Combined, G80R confirms a direct role of LOV photocycle kinetics on ZTL activity and that V48I acts in an allosteric switch to enable light-state V48I (adduct formation) to mimic the more degradation active and less GI-binding competent dark-state.

## ZTL variants alter circadian period

Given the effect of V48I, G80R and V48I:G80R on PRR5 and TOC1 degradation, these variants should have predictable effects on circadian period. Previous studies of ZTL-ox variants demonstrate a dose dependent effect of ZTL on circadian period, where high protein levels lead to short-period phenotypes progressing to arrhythmicity and ZTL-nulls having a long-period phenotype (*Somers et al., 2004*). Similarly, TOC1 overexpression strains have a long-period phenotype and TOC1-null strains have short periods (*Más et al., 2003b*; *Gendron et al., 2012*). Thus, we predict that despite overexpression the defect in degradation of TOC1 by G80R-ox should lead to WT periods. In contrast, low TOC1 levels in V48I and V48I:G80R should lead to arrhythmic phenotypes under expression levels comparable to WT-ox strains demonstrating normal periods.

Indeed examination of circadian periods in WT and mutant-ox strains confirms predicted effects of photocycle kinetics (G80R) and the signaling defect (V48I) on circadian period. All strains

**Table 3.** Period length estimates of *CCA1:LUC* activity in WT and *ZTL* variants overexpression plants. See *Figure 5—figure supplement 1* for expression levels.

| Genotype | Estimated Period length (hrs) | HA-ZTL protein abundance | *Estimated kdeg PRR5 LD (hrs−1) | *Estimated kdeg PRR5 LL (hrs−1) |
|---|---|---|---|---|
| WT | 24.36 ± 0.40 | ND | 0.3 ± 0.1 | 0.13 ± 0.1 |
| *35S:HA-ZTL* #7 | 23.69 ± 1.23 | 1 | 0.5 ± 0.1 | 0.13 ± 0.1 |
| *35S:HA-ZTL* #17 | 23.65 ± 0.72 | 1.5 ± 0.3 | ND | ND |
| *35S:HA-ZTL (V48I)* #7 | ND | 11 ± 4 | 0.8 ± 0.1 | 0.5 ± 0.2 |
| *35S:HA-ZTL (V48I)* #22 | ND | 14 ± 5 | ND | ND |
| *35S:HA-ZTL (G80R)* #22 | 24.17 ± 1.08 | 3 ± 0.5 | 0.34 ± 0.05 | 0.14 ± 0.1 |
| *35S:HA-ZTL (G80R)* #23 | 19.33 ± 1.06 | 12 ± 5 | ND | ND |
| *35S:HA-ZTL (V48I:G80R)* #45 | ND | 10 ± 3 | 0.8 ± 0.2 | 0.3 ± 0.2 |
| *35S:HA-ZTL (V48I:G80R)* #49 | ND | 7.6 ± 2 | ND | ND |

*Estimated k$_{deg}$ values were extracted by fitting *Equation S10* (below) to the PRR5 protein levels in vivo. For LD conditions, an average k is obtained by treating the system as only containing dark-state protein. Thus, the LD values are accurate as comparative terms between variants only.

containing V48I lead to an arrhythmic phenotype consistent with heightened degradation activity in these variants (*Figure 6* and *Table 3*). In contrast, G80R-ox strains harboring 3-times more ZTL protein than WT has a circadian period indistinguishable from WT and WT-ox (see *Table 3*; *35S:HA-ZTL (G80R)* #22; 24 hr, compared to WT; 24 hr and 35S:HA-ZTL #17; 24 hr). Only when protein levels exceed 10-fold higher than WT-ox is the period shortened to 19 hr (*35S:HA-ZTL (G80R)* #23) (*Figure 6* and *Table 3*). These results confirm that ZTL photocycle kinetics are coupled to selection of circadian period through PRR5/TOC1 degradation.

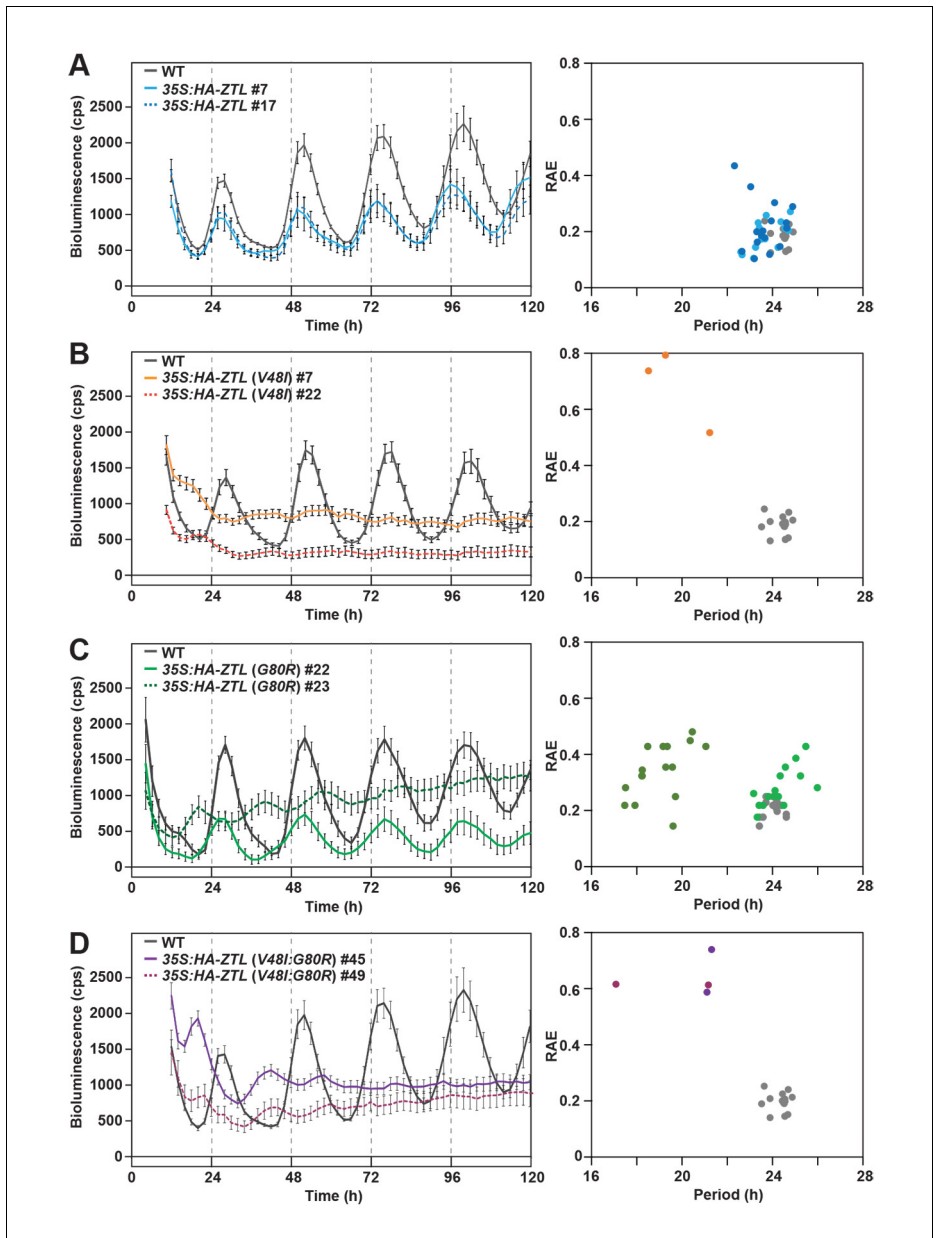

**Figure 6.** Circadian clock phenotypes of *ZTL* variant overexpressors. (A–D) *CCA1:LUC* activity was analyzed in WT, *35S:HA-ZTL* (**A**), *35S:HA-ZTL* (*V48I*) (**B**), *35S:HA-ZTL* (*G80R*) (**C**) and *35S:HA-ZTL* (*V48I:G80R*) (**D**) lines under continuous light conditions. *CCA1:LUC* traces represent the averages ±SEM of the results obtained from eight individual seedlings. Period length estimation and relative amplification errors of 16 individual measurements are shown.

## Discussion

### Insight into ZTL function in arabidopsis

In *Arabidopsis thaliana*, ZTL and FKF1 diverge in their diurnal pattern of transcription and function. Whereas, FKF1 function and transcription is specific to the day, ZTL demonstrates constitutive transcription and distinct functions during the day and night (*Song et al., 2014*). These distinct functions allow ZTL to impact photoperiodic flowering through morning specific and GI-dependent destabilization of CO, and circadian function through night specific degradation of TOC1 and PRR5 (*Más et al., 2003a*; *Song et al., 2014*; *Kiba et al., 2007*). These results suggest that both light (CO destabilization/GI interaction) and dark (TOC1/PRR5 degradation) activate ZTL for function. Such observations seem paradoxical; however, examination of our structural and kinetic results indicates ZTL may use a divergent allosteric mechanism to enable dual light/dark functions.

The ability of LOV domains to function as photoswitchable proteins is predicated on LOV proteins specifying a distinct light- and dark-state configuration. In all dark and light-state LOV structures currently in the protein data bank, the active site Gln (Q154 in ZTL) adopts a distinct buried conformation in the dark, with the NH moiety of the Gln side chain near flavin-N5 (*Halavaty and Moffat, 2007*; *Zoltowski et al., 2007*). Existing light-state structures indicate that C4a adduct formation and N5 protonation induces rotation of the Gln side chain to favor an H-bond between flavin-HN5 and the O moiety of the Gln side chain (*Vaidya et al., 2011*). In those structures, the Gln residue maintains the buried conformation and is only distinguished by the nature of H-bonds. In our ZTL structures this is not the case, rather dark-state structures have a heterogeneous orientation of Q154 that samples orientation near both the buried conformation and an exposed conformation (*Figure 3—figure supplement 1*). A heterogeneous conformation of the key signaling switch would suggest poor regulation of function under dark-state conditions. This is exactly what is observed in ZTL. ZTL retains fairly robust activity for GI interactions and TOC1/PRR5 degradation in both the light and dark, with light enhancing GI interactions and repressing TOC1/PRR5 degradation by 3–5 fold (*Kim et al., 2007*) and (*Table 3*). As such, GI-dependent function in CO destabilization is likely not a light-regulated event, but rather is a combined result of constitutive transcription of ZTL, day-specific expression of GI and poor signal amplification following light-dark interconversion due to the heterogeneous orientation of Q154. Thus, the unusual Gln orientations appear to be evolutionarily selected to permit light- and dark-state functions of ZTL and to differentiate ZTL and FKF1.

Combining our dark- and light-state structures with in vivo data allows construction of a putative signaling mechanism differentiating ZTL from other LOV proteins. Examination of the ZTL structures identifies two key residues involved in regulating ZTL allostery. The heterogeneous conformation of Q154 is permitted by the lack of a sidechain in G46 (model in *Figure 4—figure supplement 2*). Introduction of V48I, directs Q154 to a single exposed conformation in the dark. Biasing the Q154 to the exposed conformation results in disrupted interactions with GI and high TOC1/PRR5 degradation, in vivo, consistent with selecting for a distinct dark-state conformation. Further, the light-state V48I:G80R molecule indicates partial Q154 rotation to a buried conformation that is impeded by V48I. This impediment does not allow robust sampling of the buried conformation under any lighting conditions and coincides with constitutively high TOC1/PRR5 degradation in V48I containing variants. Combined we propose a putative model of ZTL signaling (*Figure 4*). ZTL retains functionality in the light and dark due to a heterogeneous orientation of Q154 permitted by G46 and V48. Differences in functionality between dark/light result from subtle biases in the orientation of Q154 between buried and exposed conformations. Based on V48I:G80R structures and activity, we propose that biasing towards the exposed conformation as the dark-state configuration (enhanced PRR5/TOC1 degradation) and biasing towards the buried conformation as the light-state configuration (enhanced GI binding).

### G46 and V48 are evolutionarily selected to differentiate ZTL and FKF1 in plants

Based on our proposed mechanism distinguishing ZTL signal transduction from other LOV proteins, one would predict evolutionary selection of G46 and V48 in ZTL proteins to permit the heterogeneous orientation of Q154. Phylogeny of LOV sequences in planta demonstrates evolutionary selection of residues G46, V48 and F156 to differentiate ZTL-like, FKF1-like and phototropin-like (LOV2)

proteins and putative signal transduction pathways (*Figure 7* and *Figure 7—figure supplement 1*). Specifically, in monocots and dicots ZTL-like proteins conserve the G46 that is necessary for selection of the exposed conformation of Q154. In FKF1 (monocots: A46 and dicots: S46), phototropins (N46) and all other structurally characterized LOV proteins, an H-bond or sterically directing residue occupies this position.

Residue selection at position 46 is coupled to V48. V48 is conserved in ZTL, but substitutions are permitted in FKF1, phototropins and fungal LOVs (*Figure 7—figure supplement 1*) (*Lokhandwala et al., 2015*; *Zoltowski et al., 2007*). The altered signaling mechanism in ZTL is

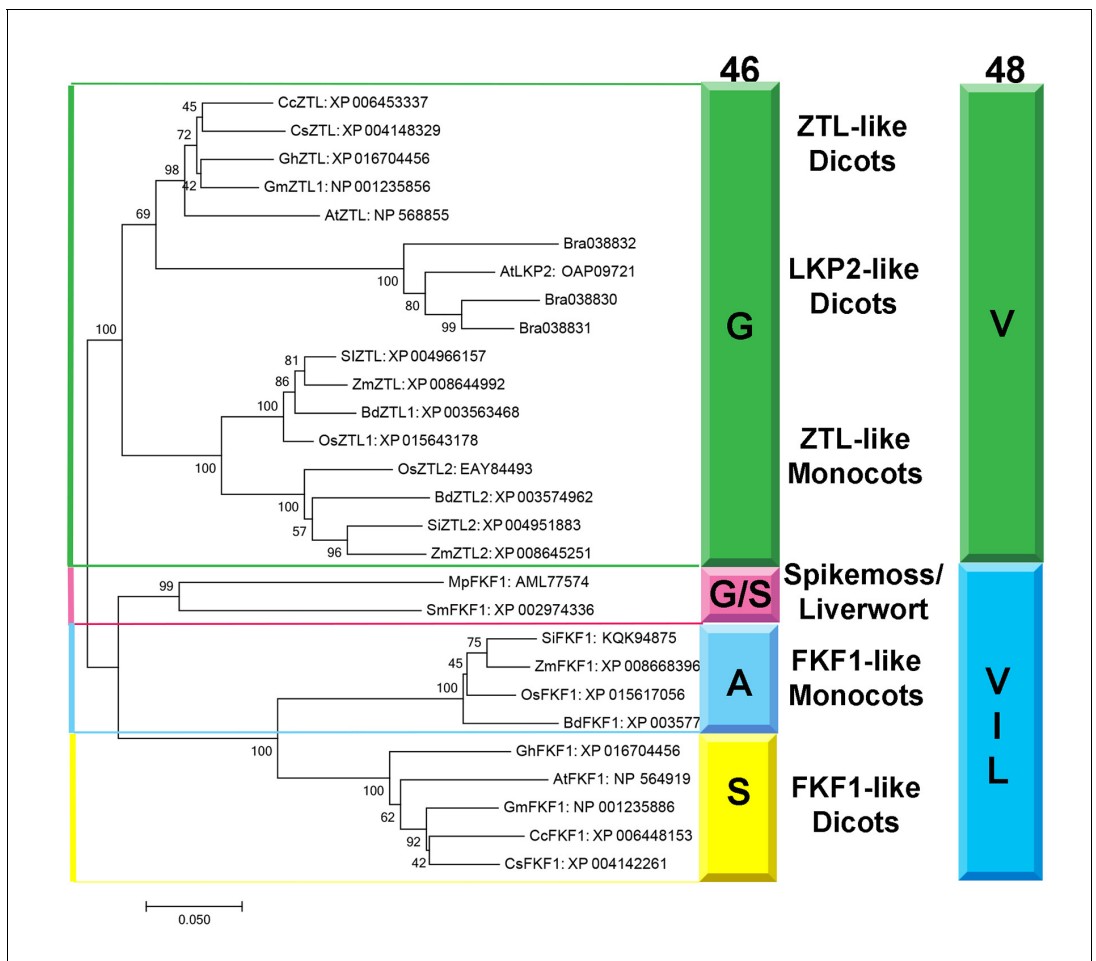

**Figure 7.** Phylogenetic Analysis of FKF1/ZTL family members in plants. Residue identity at position 46 (Colored Bar) distinguishes ZTL-like, LKP2-like and FKF1-like proteins consistent with evolutionary diversification of signaling mechanisms. LKP2 is isolated to a clade containing *Brassica rapa* members that all contain a Q154L substitution. Al ZTL members contain G46 which is necessary to promote the alternative conformation of Q154. Spikemoss and liverwort FKF1's are isolated indicating a possible intermediate function. Accession numbers for all sequences are shown after the protein name. The evolutionary history was inferred using the Minimum Evolution method (*Rzhetsky and Nei, 1992*). The percentage of replicate trees in which the associated taxa clustered together in the bootstrap test (500 replicates) are shown next to the branches (*Felsenstein, 1985*). The tree is drawn to scale, with branch lengths in the same units as those of the evolutionary distances used to infer the phylogenetic tree. The evolutionary distances were computed using the Poisson correction method (*Zuckerkandl and Pauling, 1965*) and are in the units of the number of amino acid substitutions per site. The ME tree was searched using the Close-Neighbor-Interchange (CNI) algorithm (*Nei and Kumar, 2000*) at a search level of 1. The Neighbor-joining algorithm (*Saitou and Nei, 1987*) was used to generate the initial tree. The analysis involved 28 amino acid sequences. All positions containing gaps and missing data were eliminated. There were a total of 535 positions in the final dataset. Evolutionary analyses were conducted in MEGA7 (68). (*Figure 7—figure supplement 1*).

The following figure supplement is available for figure 7:

**Figure supplement 1.** Sequence Alignment of FKF1/ZTL family members in plants.

supported by examination of LOV proteins in *Brassica rapa*. ZTL-like proteins in *Brassica* and *At*LKP2 contain a Q154L substitution, resulting in an LXF consensus sequence at that locus. The presence of a Q154L substitution is unexpected, since these substitutions abrogate blue light signaling in other LOV proteins due to an inability to undergo the Gln-flip mechanism (*Zoltowski et al., 2007*; *Nash et al., 2008*; *Nozaki et al., 2004*). In the proposed ZTL mechanism a Gln-flip is not necessitated and Q→L substitutions are permitted. In this manner, our understanding of a canonical model of LOV signal transduction is incomplete. We propose that evolutionary selections at G46 and V48 tune allostery to the Ncap through the Q154 orientation to differentiate modes of LOV signal transduction. Given the evolutionary selection of A/S residues and permission of V48 substitutions in FKF1, we propose that FKF1 functions in a manner analogous to other LOV proteins, where light activates the biological function (CO stabilization/CDF degradation) of the primarily nuclear FKF1 protein. In this manner, evolutionary selection at position 46 may dictate day functional (FKF1) and day/night functional (ZTL) differentiation, thereby implicating LOV photocycle kinetics as being imperative for proper signal transduction.

## ZTL:TOC1/PRR5:GI circuit dictates circadian period through LOV kinetics

Currently, delays in TOC1 and PRR5 degradation are explained using competition between GI and TOC1/PRR5 for the available ZTL pool (*Más et al., 2003a*; *Kim et al., 2007*; *Kiba et al., 2007*; *Fujiwara et al., 2008*). In these models, GI expression during the day both stabilizes and sequesters ZTL. In the evening, GI pools decline leading to active free ZTL. In light of our data, the competition model must be incomplete. In the competition model there should be no differences in the LL/LD characteristics in WT and G80R strains. In the competition model GI expression profiles should dictate delays and these would be unaffected by rate-altering ZTL variants. WT and G80R interact with GI to comparable levels, yet G80R demonstrates enhanced delays in PRR5 and TOC1 degradation (*Figure 5D*). These results are inconsistent with a competition model alone. In contrast, modeling PRR5 degradation using the time dependent conversion of ZTL-light to ZTL-dark can predict with reasonable certainty the extended delay in G80R and predicts distinct differences under LL conditions between the two proteins (*Figure 5—figure supplement 2*). In this manner, it is clear that ZTL photocycle kinetics are instrumental in dictating delays in PRR5/TOC1 turnover and circadian period.

Based on all these elements we propose that circadian period is regulated in a complex manner involving a ZTL:TOC1:GI circuit, whereby competitive inhibition and ZTL photocycle kinetics act in concert to dictate ZTL protein levels and ZTL activity in a circadian phase dependent manner. Adduct decay in ZTL then enhances degradation of PRR5 and TOC1, impacting circadian period through two methods: (1) Degradation of PRR5, helps diminish TOC1 levels through increased cytosolic retention and accessibility to ZTL (*Wang et al., 2010*) and (2) Degradation of cytosolic TOC1. Both factors require proper ZTL photocycle kinetics. Thus, LOV photocycle kinetics are instrumental in evolutionary selection for a 24 hr period.

# Materials and methods

## Model generation

Examination of recent mathematical models of plant circadian clocks reveals a common method of incorporating light-dark dependent degradation of PRR5 and TOC1 by ZTL. In Pokhilko *et al.*, PRR5 and TOC1 levels are treated as follows (*Pokhilko et al., 2013*):

$$\frac{dc_{p5}}{dt} = p_{10}c_{p5}^m - (m_{17} + m_{24}D) * c_{P5} \tag{S1}$$

$$\frac{dc_{TOC1}}{dt} = p_4\left(c_{TOC1}^m + n_{16}\right) - (m_6 + m_7D) * c_{TOC1} * (c_{ZTL} * p_5 + c_{ZG}) - m_8c_{TOC1} \tag{S2}$$

Where $C_x$ represent protein concentrations, $C_x^m$, mRNA levels, $p_x$, $n_x$ and $m_x$ are parameters fit to data sets. ZG represents the ZTL:GI complex. D represents darkness, where D = 1 at night and D = 0 during the day. Both equations are constructed of similar elements, a protein synthesis term defined by mRNA levels and a degradation term defined by PRR5/TOC1 protein levels, and in the

case of TOC1 the total ZTL protein pool. We note, that ZTL protein levels are not incorporated into existing models of PRR5 degradation (*De Caluwé et al., 2016*; *Pokhilko et al., 2013*) suggesting that overexpression has a weak enough effect on the overall rate of PRR5 degradation that models lacking ZTL concentration can accurately predict degradation kinetics. Although biologically such an analysis is incomplete, for modeling purposes the accuracy of these prior models suggests this assumption is reasonably valid. Therefore, they normalize the ZTL concentration to one and it does not appear in the PRR5 degradation term. Further, in the fit parameters, p5 = 1, which allows simplification as shown in *Equation S4*, where $c_{ZTL}$ now equals the total pool of ZTL protein regardless of whether free or in complex with GI.

For our purposes, ZTL rate altering variants should not perturb the transcription rates, therefore differences in PRR5/TOC1 degradation should be limited to the degradation terms. For PRR5 and TOC1 the degradation terms are as follows

$$\frac{dc_{p5}}{dt} = -(m_{17} + m_{24}D) * c_{P5} \tag{S3}$$

$$\frac{dc_{TOC1}}{dt} = -(m_6 + m_7D) * c_{TOC1}c_{ZTL} - m_8 c_{TOC1} \tag{S4}$$

The above equations do not incorporate ZTL photochemistry, rather, the term m*D to enhance the degradation rate constant under dark state conditions. In the dark, D = 1 and the degradation rate constant ($k_2$) becomes ($m_{17} + m_{24}$) or ($m_6 + m_7$) for PRR5 and TOC1 respectively. In the light, D = 0 and the rate constants ($k_1$) reduce to $m_{17}$ and $m_6$. To add ZTL photochemistry to these equations we rewrite these degradation equations in terms of light and dark-state ZTL. We also note additional complications in the TOC1 equation. The term $m_8 * c_{TOC1}$ is a non-ZTL dependent degradation term, presumably accounting for nuclear degradation of TOC1. Because of the additional complexities in TOC1 expression and non-ZTL dependent degradation we do not model TOC1 degradation in vivo, we do however show below that the rate constant for LOV adduct decay, $k_3$, will impact TOC1 in a manner analogous to PRR5.

$$\frac{dc_{p5}}{dt} = -(k_1 * c_{ZTL-L} + k_2 * c_{ZTL-D}) * c_{P5} \tag{S5}$$

$$\frac{dc_{TOC1}}{dt} = -(k_1 * c_{ZTL-L} + k_2 * c_{ZTL-D}) * c_{TOC1} - m_8 * c_{TOC1} \tag{S6}$$

Where $k_1$, $k_2$ are the rate constant for degradation in the light and dark respectively. Similarly, $c_{ZTL-L}$ and $c_{ZTL-D}$ represent the light and dark-state levels of ZTL. We also normalize the total ZTL concentration to 1. Doing so allows calculation of $c_{ZTL-L}$ and $c_{ZTL-D}$ as a function of time during the dark-phase of LD cycles, by incorporating the rate constant for adduct decay, $k_3$.

$$c_{ZTL-D} = 1 - c_{ZTL-L} \tag{S7}$$

$$c_{ZTL-L}(t) = e^{-k_3 t} \tag{S8}$$

It follows that:

$$\frac{dc_{p5}}{dt} = -[k_1 * c_{ZTL-L} + k_2 * (1 - c_{ZTL-L})] * c_{P5} \tag{S9}$$

$$\frac{dc_{p5}}{dt} = -[(k_1 - k_2) * e^{-k_3 t} + k_2] c_{P5} \tag{S10}$$

Similary:

$$\frac{dc_{TOC1}}{dt} = -[k_1 * c_{ZTL-L} + k_2 * (1 - c_{ZTL-L})] * c_{TOC1} - m_8 * c_{TOC1} \tag{S11}$$

$$\frac{dc_{TOC1}}{dt} = -\left[(k_1 - k_2) * e^{-k_3 t} + k_2\right] c_{TOC1} - m_8 * c_{TOC1} \tag{S12}$$

We note, that a compact model of clock function by De Caluwe *et al.* demonstrates that existing data can be adequately fit for both PRR5 and TOC1 levels using an analogous equation for both PRR5 and TOC1 (only the degradation term has been extracted) (*De Caluwé et al., 2016*).

$$\frac{dc_{PT}}{dt} = -(k_D D + k_L L) * c_{PT} \tag{S13}$$

Where, $k_D$, $k_L$ are the light and dark degradation rate constants respectively, and L and D reference the lighting conditions, where L = 1, D = 0 during the day and L = 0, D = 1 during the evening. $C_{PT}$ represents either the PRR5 or TOC1 level. Applying our method above to the De Caluwe *et al.* model (*Equation S13*), results in an analogous expression as *Equation S10*.

Thus, in terms of ZTL chemistry, PRR5 and TOC1 degradation should be dictated by $k_3$. Using mutants that affect only $k_3$, see *Table 1*, one should be able to accurately model PRR5 decay rates in vivo. Again we note that the expression pattern of TOC1 and complex regulation of TOC1 mRNA complicates TOC1 levels during the circadian cycle. For these reasons, we use the simpler PRR5 degradation data to mathematically test our model above and use qualitative differences in degradation patterns to examine effects of $k_3$ on TOC1.

## Parameter estimation for *Figure 1B*

To predict how $k_3$ may affect delays in PRR5 degradation we required estimates of $k_1$ and $k_2$. Examining Pokhilko *et al.* and De Caluwe *et al.* provides reasonable bounds for parameter estimation. First, PRR5 models suggest that PRR5 can be accurately modeled excluding differences in ZTL expression levels (*De Caluwé et al., 2016*; *Pokhilko et al., 2013*), this is consistent with our data showing that WT, WT-ox and G80R-ox all have the same decay rate under LL conditions despite differences in protein levels. Second, in De Caluwe *et al.* the maximum degradation rate constant for PRR5 degradation was fit as 0.78 hr$^{-1}$. Similarly, in Pokhilko *et al.* a max value of 0.5 hr$^{-1}$ was obtained for PRR5 and a $k_{max}$ of 0.8 hr$^{-1}$ for TOC1 by combining all degradation terms. Given the similarity of these maximum values we chose a $k_{max}$ for the dark-state of 0.8 hr$^{-1}$. The lowest fit rate constant in mathematical models is $m_6$ (0.2 hr$^{-1}$) for TOC1 degradation under light-state conditions. Thus, we used 0.2 hr$^{-1}$ as an estimate of PRR5/TOC1 degradation in the light.

Data was then simulated by numerically solving *Equation S9* in matlab using the following parameters and plotted in *Figure 1B*.

$k_1 = 0.2$ hr$^{-1}$

$k_2 = 0.8$ hr$^{-1}$

$k_3 = 0.7$ hr$^{-1}$ (WT), 0.15 hr$^{-1}$ (G80R), 0.09 hr$^{-1}$ (V48I), 0.05 hr$^{-1}$ (G46S:G80R), 0.02 hr$^{-1}$ (V48I:G80R)

PRR5(0)=1

Values for $k_3$ are derived from experimental values for the adduct decay time constant ($\tau$) present in *Table 1*, where $k_3 = 1/\tau$.

## Fitting model to experimental data

To estimate the relative accuracy of our model depicted in *Equation S10*, we extracted improved estimates of $k_1$ and $k_2$ from the experimental data shown in *Figure 5D*.

Our structural data indicates that V48I mimics the dark-state regardless of lighting conditions, thus, V48I degradation kinetics under LD serves as a reasonable estimate for $k_2$, the dark-state rate constant. We note, that our experimental value for V48I (0.8 hr$^{-1}$) is identical, to the $k_{max}$ values present in both Pokhilko *et al.* and De Caluwe *et al*, which should reference the more active dark-state value.

Under LL conditions we observe a rate constant for PRR5 degradation of 0.13–0.14 hr$^{-1}$ for WT, WT-ox and G80R. Under our lighting conditions (broad spectrum; 40–50 µmol m$^{-2}$s$^{-1}$), the light-state should be near saturated, and thus in the absence of allosteric effects should report the light-state degradation rate constant. We use 0.14 hr$^{-1}$ then as an estimate for $k_1$ the light-state rate constant.

We note the similarity of values between the three strains despite different expression levels. These results suggest that the models by Pokhilko *et al.* and De Caluwe *et al.* demonstrating a weak dependence of PRR5 degradation on ZTL expression levels is reasonably valid under our conditions. Further, it provides a reasonable estimate for $k_1$ the light-state degradation term.

Thus, to test the accuracy of our model we numerically solved *Equation S9* in matlab using:

$k_1$ = 0.14 hr$^{-1}$

$k_2$ = 0.8 hr$^{-1}$

$k_3$ = 0.7 hr$^{-1}$ (WT), 0.15 hr$^{-1}$ (G80R)

Initial PRR5 levels were taken from *Figure 5D* at t = 12 hr (initial dark).

Values for $k_3$ are derived from experimental values for the adduct decay time constant ($\tau$) present in *Table 1*, where $k_3 = 1/\tau$.

Results were plotted in *Figure 5—figure supplement 2*.

## Cloning and purification

ZEITLUPE construct ZTL-S composed of residues 29–165 were cloned into both p-His and pGST parallel vectors using NcoI and XhoI restriction sites. Proteins were purified as reported previously (*Pudasaini and Zoltowski, 2013*). These DNA sequences were verified by GENEWIZ sequencing service. The plasmids were then isolated and tested for protein expression. All constructs were transformed into *E. coli* (JM109 or JM109DE3) cells and grown overnight at 37°C as starter culture till O.D$_{600}$ ~0.6–0.7. The rich culture was then transferred into 1.0 L of LB media for large-scale expression. These cultures were grown for 2–3 hr at 37°C till O.D$_{600}$ ~0.5–0.6 then the temperature was lowered to 18°C. The culture was then induced with 200 μM Isopropyl-$\beta$-D-thio-galactopyranoside (IPTG) after 1 hr incubation at 18°C. After induction, the cells were grown at 18°C for about 18–20 hr. Finally, the cells were centrifuged at 4000 rpm to collect and store the cells in stabilizing buffer (50 mM Tris pH 7 or 50 mM Hepes pH 8 with 100 mM and 10% glycerol). The harvested cell-pellets were stored for later use at −80°C.

We note that ZTL is difficult to express and purify from *E. coli* as most WT proteins are confined to inclusion bodies. Typical experiments required cell pellets from 18 L of cells. Such difficulties make studies of variants such as G46S difficult in the absence of G80R. The G80R variant, likely due to the stabilizing effects of the R80-F84 π-cation interaction (*Figure 2B*), enhances protein yields in *E. coli* substantially. This allows access to G46S:G80R, despite intractable yields of the isolated G46S variant.

ZTL-S was purified using affinity chromatography followed by size exclusion chromatography. Prior to purification the cells were lysed by sonication at 4°C. After sonication the protein solution was centrifuged at 18 k rpm at 4°C for 60 min. The supernatant was then purified using Ni-NTA or GST affinity columns. 6xHis and GST tags were cleaved using 6His-TEV-protease followed by an additional round of Ni-NTA affinity chromatography to remove the 6His tag and 6His-TEV-protease. The final eluted protein was subjected to Fast Protein Liquid Chromatography (FPLC) using a Hiload Superdex 200 16/60 gel filtration column equilibrated with stabilizing buffer.

## Size exclusion chromatography

Solution characterization of purified proteins was done using a Superdex 200 10/300 analytical column (GE Lifesciences). Protein concentrations were determined using the absorbance at 450 nm (ext. coefficient 12,500). Apparent molecular weights were calculated by comparing the elution volume of known standards (sweet potato $\beta$-amylase (200 kDa)−12.4 ml; yeast alcohol dehydrogenase (150 kDa)−13.31 ml; bovine albumin (66 kDa)−14.61 ml; carbonic anhydrase (29 kDa)−17.03 ml; horse heart cyctochrome c (12.4 kDa)−18.32 ml)) (Sigma Aldrich). Absolute molecular weights of ZTL 16–165, ZTL 29–165 and FKF1 28–174 were determined by subjecting samples to refractive index and light-scattering detectors on a Wyatt Minidawn light-scattering instrument following a Superdex 200 10/300 analytical column. MW's were determined using ASTRA software from Wyatt Technologies (Santa Barbara, California). All SEC and multi-angle light scattering experiments were conducted in stabilizing buffer (see above).

## Mutagenesis

Site specific protein variants of ZTL-S constructs were obtained using the quickchange protocol. Following PCR amplification samples were treated with 1 µL of *DpnI* and incubated at 37°C for 2.5–3 hr to cleave the methylated template DNA. A single colony of DH5α *E. coli* was grown at 37°C overnight and plasmid DNA was isolated and verified by DNA sequencing (Genewiz). Rate altering variants of ZTL were expressed, purified and characterized using the method described above.

## Structural analysis

ZTL-S and its variants were initially screened with Hampton Screens (HR2-110 and HR2-112) via the hanging drop methods using 1.5 µL well solution with 1.5 µL of ZTL-S at various concentration range of 5–10 mg/ml. Optimum crystallization conditions for WT (0.1 M Tris pH 8.5, 0.1 M Magnesium Chloride hexahydrate, 28% w/v PEG 4k), G80R (0.1 M Tris pH 8.5, 0.2 M Magnesium Chloride hexahydrate, 30% w/v PEG 4k), V48I:G80R (0.1 M Tris ph 8.5, 0.2 M Sodium Acetate trihydrate, 30% w/v PEG 4K). Protein for crystallographic studies was purified in the same stabilizing buffer.

Light state crystals of V48I:G80R were obtained as follows. Prior to setting screens V48I:G80R protein was exposed to a broad spectrum white flood light (150 W), while on ice for two minutes. Saturation of the light-state was confirmed by UV-vis spectra analysis, by verifying disappearance of the 450 and 478 nm absorption bands characteristic of the dark-state protein. The light-state protein was then crystallized directly using the hanging drop method outlined above. Crystals appeared within 24 hr, and trays were exposed to white light once a day to maintain population of the light-state species.

Diffraction data was collected at the F1 beamline at the Cornell High-Energy Synchrotron Source (CHESS). Data for WT and all variants was collected at 100 K. The following cryoprotectants were added: V48I:G80R dark (25% Glycerol v/v), V48I:G80R light (25% Glycerol v/v), G80R (25% ethylene glycol v/v), and WT (25% ethylene glycol v/v). Data was scaled and reduced in HKL2000 (*Otwinowski and Minor, 1997*) (see *Equation S2* for refinement statistics). The initial WT structure was solved using molecular replacement in PHASER (*McCoy et al., 2007*) and PHENIX (*Adams et al., 2010*) with the LOV1 domain of *Arabidopsis* phototropin 2 (PDBID 2Z6D) as a search model. Structures of ZTL variants were solved using the same method with WT ZTL as the search model. Rebuild cycles were completed in COOT (*Emsley and Cowtan, 2004*) and refinement with REFMAC5 (*Murshudov et al., 1997*) and PHENIX (*Adams et al., 2010*). All ZTL structures contain four molecules per asymmetric unit that is composed of two anti-parallel LOV dimers. Residues 29–31 are not visible in the electron density in any molecule. In several molecules residues 29–43 and 164–165 are unable to be resolved and have not been built. For light-state structures clear density is observed for the adduct state in two of four molecules. Although electron density suggests an adduct for the remaining two molecules, we do not model them with an adduct and the reduced density likely reflects reduction by x-rays during data collection as has been observed previously (*Zoltowski et al., 2007*).

## UV-absorption spectroscopy and kinetics

Purified protein fractions were concentrated to 30–60 µM for UV-Vis absorption spectroscopy measurements and kinetics. Samples were exposed to a broad spectrum white flood light (150 W), while on ice for two minutes. An Agilent UV3600 spectrophotometer was used to characterize the absorption spectra of all constructs in both light and dark states. The light state peak of 378 nm (ext. coefficient 8500) and dark state peak 450 nm (ext. coefficient 12,500) were used to estimate the protein concentrations for experiments.

Photocycle recovery kinetics were analyzed by measuring the absorption at 450 and 478 nm as a function of time. Spectra were collected at intervals ranging from 100 seconds-2 hours to ensure minimal repopulation of the light-state by the probe source. Time intervals were chosen to maintain approximately 10–20 measurements per half life. Kinetic traces at 450 and 478 nm were then fit with a monoexponential decay of the form $y = y_0 + A*e^{-k*x}$. The rate constant k and time constant (1/k) were abstracted. Results are presented in *Table 1* as the average of three independent measurements.

## Plant materials and growth conditions

The Columbia-0 (Col-0) plant that possesses *CCA:LUC* reporter was previously described (*Pruneda-Paz et al., 2009*). To generate overexpressors of HA-ZTL, HA-ZTL (V48I), HA-ZTL(G80R) and HA-ZTL (V48IG80R), the nucleotide sequences encoding HA tag was incorporated into the *ZTL* forward primer (5'-CACCATG<u>TACCCATACGATGTTCCAGATTACGCT</u>GAGTGGGACAGTGGTTC-3', the underline indicates the sequences that encodes HA tag). Amino acid substitutions on ZTL coding region were generated by using megaprimer-based PCR amplification method (*Burke and Barik, 2003*). The primers used for generating the mutated *ZTL* coding sequence are followings; ZTL (V48I) _R (5'- AACGGCATCAGTAAC<u>AAT</u>GAATCCACAAGGCGC-3'), ZTL (G80R)_R (5'- CAAGAAGCGG-CAATT<u>TCG</u>TCCGAGAACTTCCTC-3'), ZTL_R (5'- TTACGTGAGATAGCTCGCTA-3'). Amplified PCR fragments were cloned into pENTR/D-TOPO vector (Invitrogen). After verifying sequences, *HA-ZTL* and mutated *HA-ZTL* coding regions were transferred into pB7WG2 or pH7WG2 binary vector (*Karimi et al., 2002*) using LR Clonase II enzyme mix (Invitrogen) to generate *35S:HA-ZTL*, *35S:HA-ZTL (V48I)*, *35S:HA-ZTL (G80R)* and *35S:HA-ZTL (V48I:G80R)*. The binary vectors were introduced into the *CCA1:LUC* plants by conventional *Agrobacterium*-mediated transformation method. The T3 generations of transgenic lines in which the expression levels of ZTL variant mRNAs were similar were chosen for the circadian analysis. The plants were grown on soil or Linsmaier and Skoog (LS) media (Caisson) in plant incubator (Percival Scientific; Perry, Iowa) set at 22°C under full-spectrum white fluorescent light (70–90 µmol m$^{-2}$s$^{-1}$: F017/950/24', Octron Osram Sylvania) in long days (16 hr light/8 hr dark).

## Bioluminescence imaging

Bioluminescence Imaging and analysis were performed as previously described with minor modifications (*Fenske et al., 2015*). Seedlings were grown on LS media in the plant incubator (Percival Scientific) in 12 hr light/12 hr dark photoperiod for 10 days before being transferred to continuous light (40–50 µmol m$^{-2}$s$^{-1}$) conditions. 9-day-old seedlings were pretreated with 5 mM D-luciferin (Biosynth) in 0.01% Triton X-100 solution, and incubated one day before imaging. The bioluminescence generated from the *CCA1:LUC* reporter was imaged for 15 min at every 2 hr using NightOwl system (Berthold; Germany) and analyzed using IndiGO software (Berthold). Period length estimation was performed using fast Fourier Transform-Nonlinear Least Squares (FFT-NLLS) analysis in the Biological Rhythms Analysis Software System (BRASS) (http://millar. bio.ed.ac.uk/PEBrown/BRASS/BrassPage. htm).

## RNA isolation and gene expression analysis

For gene expression analyses, 14-day-old seedlings grown on LS agar plates were harvested at ZT16 and used for RNA extraction using illustra RNAspin Mini kit (GE Healthcare; Chicago, Illinois). 2 µg of RNA was reverse-transcribed using iScript cDNA synthesis kit (Bio-Rad; Hercules, California). The cDNA was diluted by adding 40 µL of water, and 2 µL of cDNA was used for quantitative polymerase chain reaction (q-PCR) using MyiQ real-time thermal cycler (Bio-Rad). *IPP2* expression was used as an internal control to normalize cDNA amount. Primers and PCR conditions used for *IPP2*, *PRR5*, *TOC1* and *ZTL* amplification were previously described (*Baudry et al., 2010*). Expression of *ZTL*, *PRR5 and TOC1* was calculated from three biological replicates.

## Western blots

To analyze the expression levels of HA fused ZTL protein, endogenous PRR5, and TOC1 proteins, seedlings were grown in 12 hr light/12 hr dark conditions for 14 days. Total protein was extracted using extraction buffer [50 mM Na-P pH7.5, 150 mM KCl, 1 mM DTT, 1 mM EDTA, 0.05% Sodium deoxycholate, 0.1% SDS, 50 µM MG-132, Protease inhibitor cocktail (Pierce)}. Protein was separated in 9% SDS-PAGE gels and transferred to Nitrocellulose membrane (Bio-Rad). HA-ZTL and Actin were detected using anti-HA (3F10, Roche) and anti-actin (C4, Millipore) antibodies, respectively. Western procedure for detecting TOC1 and PRR5 proteins was previously described (*Baudry et al., 2010*). For protein quantification, western blot images were analyzed using Image J (*Schneider et al., 2012*).

## Co-immunoprecipitation assays

For *Figure 4—figure supplement 1A*, *Agrobacteria* containing both GI and ZTL variants were coinfiltrated into 4-week-old *N. benthamiana* leaves. The infiltrated plants were incubated under LD for two days and transferred to continuous light or dark with additional 24 hr incubation. Co-IP was performed according to *Fujiwara et al. (2008)*. The immuno-complexes were resuspended in SDS sample buffer and heated briefly. GFP antibody (Invitrogen, A11120) was used for immunoprecipitation of GI-GFP protein. ZTL variants were detected by western blotting with HA antibody. For *Figure 4—figure supplement 1B*, *Agrobacterium* harboring each overexpression construct was mixed according to combinations indicated and was infiltrated into 3-week-old *N. benthamiana* leaves and incubated in LD and either light or darkness as described above. Sample preparation, the IP buffer condition, the IP method and immunoblot procedure were described previously (*Song et al., 2012*).

## Acknowledgements

This work is based on research conducted at the Cornell High Energy Synchrotron Source (CHESS). CHESS is supported by the NSF and NIH/NIGMS via NSF award DMN-0936384, and by NIGMS award GM-103485. We thank the Herman Frasch Foundation (739-HF12 to BDZ) the NIH (R15GM109282 to BDZ, R01GM079712 to TI and R01GM093285 to DES), Next-Generation Bio-Green 21 Program (SSAC, PJ01117501 to YHS and PJ01117502 to TI, Rural Development Administration, Republic of Korea) and NSF (MCB-1613643 to BDZ) for funding. We thank Elaine Tobin for generously providing TOC1 antibody and Robert Green for experimental assistance.

## Additional information

### Funding

| Funder | Grant reference number | Author |
| --- | --- | --- |
| National Institutes of Health | R15GM109282 | Brian D Zoltowski |
| Herman Frasch Foundation for Chemical Research | 739-HF12 | Brian D Zoltowski |
| Rural Development Administration | PJ011175 | Young Hun Song Takato Imaizumi |
| National Science Foundation | MCB 1613643 | Brian D Zoltowski |
| National Institutes of Health | R01GM079712 | Takato Imaizumi |
| National Institutes of Health | R01GM093285 | David E Somers |

The funders had no role in study design, data collection and interpretation, or the decision to submit the work for publication.

### Author contributions

AP, Conceptualization, Data curation, Formal analysis, Methodology, Writing—original draft; JSS, YHS, Data curation, Formal analysis, Methodology, Writing—original draft; HS, Data curation, Formal analysis, Methodology; TK, Resources, Methodology; DES, Formal analysis, Supervision, Funding acquisition, Methodology, Writing—original draft, Writing—review and editing; TI, Conceptualization, Formal analysis, Supervision, Funding acquisition, Methodology, Writing—original draft, Writing—review and editing; BDZ, Conceptualization, Data curation, Formal analysis, Supervision, Funding acquisition, Methodology, Writing—original draft, Project administration, Writing—review and editing

### Author ORCIDs

Brian D Zoltowski, http://orcid.org/0000-0001-6749-0743

## Additional files

### Major datasets

The following datasets were generated:

| Author(s) | Year | Dataset title | Dataset URL | Database, license, and accessibility information |
|---|---|---|---|---|
| Pudasaini A, Zoltowski BD | 2017 | Structure and kinetics of the LOV domain of ZEITLUPE determine its circadian function in Arabidopsis | http://www.rcsb.org/pdb/explore/explore.do?structureId=5SVG | Publicly available at the RCSB Protein Data Bank (accession no: 5SVG) |
| Pudasaini A, Zoltowski BD | 2017 | Structure and kinetics of the LOV domain of ZEITLUPE determine its circadian function in Arabidopsis | http://www.rcsb.org/pdb/explore/explore.do?structureId=5SVU | Publicly available at the RCSB Protein Data Bank (accession no: 5SVU) |
| Pudasaini A, Zoltowski BD | 2017 | Structure and kinetics of the LOV domain of ZEITLUPE determine its circadian function in Arabidopsis | http://www.rcsb.org/pdb/explore/explore.do?structureId=5SVV | Publicly available at the RCSB Protein Data Bank (accession no: 5SVV) |
| Pudasaini A, Zoltowski BD | 2017 | Light-state Structure of Arabidopsis Thaliana Zeitlupe | http://www.rcsb.org/pdb/explore/explore.do?structureId=5SVW | Publicly available at the RCSB Protein Data Bank (accession no: 5SVW) |

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
