## [Decision Letter]

Thank you for submitting your article "Kinetics of the LOV domain of ZEITLUPE determine its circadian function in *Arabidopsis*" for consideration by *eLife*. Your article has been favorably evaluated by Detlef Weigel (Senior Editor) and three reviewers, one of whom, Richard Amasino (Reviewer #1), is a member of our Board of Reviewing Editors.

The reviewers have discussed the reviews with one another and the Reviewing Editor has drafted this decision to help you prepare a revised submission

Our decision is that the manuscript requires revision in order to be potentially suitable for publication in *eLife*. Below are the specific issues that should be addressed in a revision. We also note that the overall clarity and quality of writing should be improved.

From the Introduction:

There is little background about the roles of the clock proteins TOC, PPR5 and GI within the plant circadian clock and how their ZTL interactions/stability/signaling affect clock function. More information here would help the reader to understand how ZTL kinetics and TOC/PRR5 degradation could correlate with circadian period (Results, second paragraph). If manuscript length is an issue, the background description regarding LOV chemistry and allostery can be shortened without dropping content by writing some parts in a more precise/less redundant manner. A supplemental figure illustrating the roles of TOC, PPR5, GI, ZTL, FKF1 in the plant circadian clock (or putting Figure 1 into clock context) might also help.

The first paragraph of the Results should be moved from Results to the Introduction – perhaps after the fifth paragraph, where you describe consequences of FKF1-GI complex formation on CDF1 and CO degradation, but do not say anything about consequences of ZTL-GI interactions.

From the Results:

Second paragraph:"… a long photocycle would lead to increasing delays in PRR5 and TOC levels leading to progressively longer circadian phenotypes". It is more precise to state "TOC degradation" instead of "TOC levels". Also, the authors should explain or at least provide a reference for the connection between PRR5 and TOC levels and circadian period length.

Subsection “LOV dimerization”: "ZTL/FKF1 dimer", this is somewhat ambiguous, as far as we understand that you study ZTL dimers, FKF1 dimers, but no ZTL/FKF1 heterodimers – please clarify.

Discussion of the G46S mutant (Results, fourth paragraph; subsection “V48I disrupts ZTL:GI interactions”, first paragraph; subsection “ZTL variants alter targeted degradation of clock components”, first paragraph; Figure 1—figure supplement 2; Figure 4—figure supplement 1): In the subsection “V48I disrupts ZTL:GI interactions” G46S is reported to "enhance GI complex formation in both the dark and light states". In the same subsection the authors reference Kim WY et al., Nature 2007 for "strong GI-interactions of the G46S mutant". However, Kim et al. describe a non-detectable GI interaction for a ZTL(G46E) variant in Co-IP as well as a slower migration, therefore indicating potentially aberrant folding of full-length ZTL(G46E) in plant material. The authors should discuss the opposite effect of the G46E mutation (Kim et al., 2007) and the G46S mutation (this study) on GI interaction.

In the subsection “ZTL variants alter targeted degradation of clock components”, the authors state that the G46S mutant is not soluble and could therefore not be purified, crystallized or used for UV/Vis kinetics. Hence, mutation of Gly46 (to S or E) is critical for ZTL folding. Could the slower dark recovery kinetics of the ZTL(G46S/G80R) double mutant (time constant 21 h compared to 1.4 h for wt ZTL and 6.6 h for G80R ZTL) be a consequence of reduced protein stability or aberant folding? The authors should assess protein stability and folding of the ZTL mutants (in particular G46S/G80R) e.g. by thermofluor or CD spectroscopy to validate the kinetics and exclude that potential conformation defects affect GI-binding (Figure 4—figure supplement 1).

There are distinct migration properties of ZTL mutants (Figure 5) on SDS-PAGE. G46S, G46S:G80R and V48I:G80R migrate differently from the WT, V48I and G80R. Do you have a hypothesis as to the basis of these migration differences?

Can the ZTL-LOV crystal structure explain how the added G80R mutation stabilizes the G80R/G46S mutant compared to the single G46S. It is not intuitive because G80 is on the surface and not close to G46 (and V48). This point is interesting to address, because the correlated G46S, G80R exchange may be a stabilizing co-evolution in the FKF1-LOV domain. Moreover, G80 is only Gly in *A. thaliana* ZTL, but replaced by R/Q in all other ZTL/FKF1/Phot-LOV proteins aligned in Figure 7.

From the Discussion:

There are interesting aspects of your work that should be discussed such as:

1) Does the crystal structure provide any insights on the co-evolution of G80R and G46S/V48I in FKF1? G80 is Gly only in At-ZTL, this position is R/Q in all other proteins as well as in ZTLs of other species (aligned in Figure 7).

2) Ncap mutations G46E (Kim et al., 2007) and V48I weaken GI-binding, whereas G46S strengthens GI- binding. This difference, in particular between G46S and G46E should be discussed.

3) The conformation of Q154 in the dark-state structures of wt and G80R are heterogeneous (Figure 3). Does this mean that both the exposed and the buried conformation are present in these structures? Or is it a mixture of different "exposed" conformations, as suggested by the dark state model for wt/G80R ZTL shown in Figure 4 and in the first paragraph of the subsection “ZTL:TOC1/PRR5:GI circuit dictates circadian period through LOV kinetics”? Also, how is the wt/G80R light-state model shown in Figure 4 obtained, since only dark-state structures are available? Some more discussion is needed here. It would be helpful to show a superposition of the wt and G80R dark state Q154 with the V48I/G80R dark- and light-state structures to i) classify the heterogeneous Q154 conformations as exposed or buried and ii) to illustrate the impact of the V48I mutation on the Q154 position. It would also help to describe the heterogeneous G154 conformations of wt/G80R as "exposed/buried" already in the Results section, e.g. the second paragraph of the subsection “N-terminal CGF motif defines a locus of signal transduction that differentiates ZTL from known LOV signaling mechanisms”.

Subsection “Insight into LOV allostery and conservation in plants”, first paragraph: Please correlate buried/exposed Gln conformation with light/dark state.

Subsection “Insight into LOV allostery and conservation in plants”, second paragraph: "In ZTL, light activation leads to rotation of Q154 to a buried conformation observed in all other LOV structures". "all other LOV structures" implies "all light- and dark state structures of LOV domains that have ever been solved". The authors should be more specific here.

Your results address the difference in light-regulated target degradation between ZTL and FKF1: FKF1 promotes CDF degradation in the light while ZTL promotes TOC1/PRR5 degradation in the dark. However, in the Discussion it would be helpful to the readers to elaborate on the role of FKF1 in CO stabilization which also occurs in the light as noted in Song et al., PNAS 2014 (and some of the authors of this paper are authors on Song et al., 2014). That FKF1 is involved in stabilization and degradation may be a bit of a mystery, but it is important to note this as a discussion point and, if the structure-function data from this current work lead to any hypotheses as to how light-activated FKF1 stabilizes CO, that would be interesting and valuable for readers.

From Figures/Tables:

Figure 1—figure supplement 2: It would be easier for the reader to understand this figure if you annotate the ZTL mutants within the graphs and mention the time constants within the figure legend or to refer to Table 1. Figure 1—figure supplement 2: for base catalyses by imidazole, the authors should refer to Pudasaini and Zoltowski, 2013 to allow the reader to better assess the imidazole effect on ZTL.

Figure 2—figure supplement 1: Why was the ZTL (16-165) construct used for MALS instead of ZTL (29-165), which was used for crystallization (Materials and methods) and kinetics (Table 1)? Which constructs were used for analytical gel filtration studies (Figure 2—figure supplement 1, Figure 2, Figure 2—figure supplement 2)?

Figure 2, Figure 2—figure supplement 1/C, Figure 2—figure supplement 2: For analytical gel filtration studies, please state which SEC columns were used (e.g. S200/S75 10/300, 16/60 etc.) and add elution volumes of standard proteins for reference.

Figure 2 legend: MALS in Figure 2—figure supplement 1, not Figure S3.

Figure 2—figure supplement 2: why does light-state ZTL-R95A show a lower 280 absorbance than dark-state ZTL-R95A at equal concentration? One possible reason is that light-exposure leads to partial aggregation.

Figure 3: Put a gap between 3B and 3D. It is really difficult to distinguish the two panels as it is. Also, please label and show sticks for V48 and G46 in Figure 3 and/or 3B to show their relative location to Q154, FMN and C82.

Figure 3 legend: with sulfur-pi and pi-pi interactions you probably refer to C45 and F47. It would be helpful for readers to understand your work to explicitly mention "C45" and "F47" as involved residues.

Figure 4: Q154, not Q145.

Figure 4 State clearly in the legend if AsLOV is a dark- or light-state structure? Figure 4: There is no density for FMN. Please explain this.

Figure 4: the legend not clear as to what is the difference between these two panels. It would be better to describe 4B and 4C separately and be clear on how the two panels are related. Also, FMN in Figure 4 looks a bit distorted. Please comment on this.

Figure 4: for WT and G80R only dark-state structures are available. On what basis is the model for the WT/G80R light state developed?

Figure 4 legend: "Activation of ZTL at dusk.…". It would be more accurate to state that "activation" means dark-recovery of ZTL, i.e. generation of the dark state of ZTL.

Table 2: For easier understanding, please define D = dark, L = light and label WT and G80R structures as "dark". Please also add the number of Ramachandran outliers.

Table 2 and Results, end of fourth paragraph: according to Table 2 the V48I:G80R mutant was crystallized in the dark state (2.1 A resolution) and in the light state (2.3 A resolution). In line 172-174 the authors only mention light-state crystals of V48I:G80R and say that the light state structure is at 2.1 A (2.3 A would be correct for the light-state structure). Please correct this discrepancy.

Table 2: The data for light-state V48I:G80R are only 87% complete. Was there a radiation damage or an adduct cleavage problem during data collection? Please comment.

General Comment:

Standardize the amino acid nomenclature (1 or 3 letter code?). In most of the text it is the 1 letter code but sometimes the authors use 3 letters; e.g. in the first paragraph of the subsection “Insight into LOV allostery and conservation in plants”.

[Editors' note: further revisions were requested prior to acceptance, as described below.]

Thank you for resubmitting your work entitled "Kinetics of the LOV domain of ZEITLUPE determine its circadian function in *Arabidopsis*" for further consideration at *eLife*. Your revised article has been favorably evaluated by Detlef Weigel (Senior Editor), a Reviewing Editor, and two reviewers.

The manuscript has been improved but there are some remaining issues that need to be addressed before acceptance, as outlined below:

*Reviewer #2:*

My comments were primarily about the presentation of the work while the Xray crystallography expert had very specific requests about the work and experiments-interpretations.

In my opinion the new presentation is much improved and I don't have additional major comments or requests.

*Reviewer #3:*

The authors have adequately responded to my issues and submitted a significantly improved manuscript in content and clarity.

I do however regret that my issues regarding the Materials and methods section have apparently not been transmitted to the authors. I would like these parts to be improved before publication, as some method descriptions are too inaccurate or missing. Here, I recapitulate the most important requested changes in Materials and methods, now using line numbers of the revised manuscript:

Subsection "Structural analyses":

How were the crystals for the light state structure of V48I:G80R-ZTL obtained, i.e. what illumination conditions were used to crystallize the light state?

Subsection "Structural analyses", second paragraph: Cryo is only mentioned for V48I dark and light, not for WT and G80R ZTL.

Subsection "Structural analyses", last paragraph: The authors say that "WT ZTL was used as search model", presumably for mutant structures. But how was the crystal structure of WT ZTL solved, i.e. which search model was used for MR to get the ZTL WT structure?

Subsection "Structural analyses": The authors should explicitly mention that the crystals contain 4 molecules per asymmetric unit. The authors should also describe the final asymmetric unit content, including which residues are undefined/were not build in the model.

Subsection "Size exclusion chromatography": I appreciate that the authors added a Materials and methods for analytical SEC and MALS in their revised manuscript. Herein, the buffer composition used for analytical SEC and MALS should also be specified.

Subsection "Fitting model to experimental data", third paragraph: "under our lighting conditions" is very imprecise. Please provide a protocol/light source (wavelength/intensity) for ZTL light excitation.

Subsection "Model generation":

Please provide information about the definition and calculation of the time constant k3 that characterizes ZTL dark recovery kinetics. K3 is not introduced in the subsection "Model generation". Here, k3 is first mentioned in the second paragraph without explanation, then used in equation S8, S10, S12 without definition, and it is not explained how the k3 values at the end of the subsection “Parameter estimation for Figure 1” are defined and how they were obtained.

Subsection "UV absorption spectroscopy and kinetics": Please provide a illumination protocol (light source/wavelength/intensity) for ZTL light excitation.

Subsection "UV absorption spectroscopy and kinetics": The sentence "Kinetic traces were then fit with a monoexponential decay and kinetic parameters were abstracted." is too imprecise. Please provide a more detailed description of how the decay time constants are calculated from the absorption kinetics and provide the equation used for curve fitting. Which wavelength curve is used for fitting, 450 nm or 478 nm?

---

## [Author Response]

*Our decision is that the manuscript requires revision in order to be potentially suitable for publication in eLife. Below are the specific issues that should be addressed in a revision. We also note that the overall clarity and quality of writing should be improved.*

*From the Introduction:*

*There is little background about the roles of the clock proteins TOC, PPR5 and GI within the plant circadian clock and how their ZTL interactions/stability/signaling affect clock function. More information here would help the reader to understand how ZTL kinetics and TOC/PRR5 degradation could correlate with circadian period (Results, second paragraph). If manuscript length is an issue, the background description regarding LOV chemistry and allostery can be shortened without dropping content by writing some parts in a more precise/less redundant manner. A supplemental figure illustrating the roles of TOC, PPR5, GI, ZTL, FKF1 in the plant circadian clock (or putting Figure 1 into clock context) might also help.*

We agree with the reviewer that the manuscript would benefit from increased discussion of the role of ZTL within the circadian clock. In light of this recommendation as well as others requesting more commentary on the divergent and synergistic roles of ZTL and FKF1 we have reworked both the Introduction and Discussion to better highlight these aspects. We feel that this greatly strengthens the manuscript by placing more emphasis on the evolutionary divergence of ZTL and FKF1 to allow differential function in circadian and photoperiodic clocks.

By emphasizing the plant biology more, we are also easily able to address the reviewer request for more commentary in the discussion on evolutionary selection of G46 and V48.

We have also added an additional figure (Figure 1—figure supplement 2) summarizing the main functions of ZTL in the light and dark.

*The first paragraph of the Results should be moved from Results to the Introduction – perhaps after the fifth paragraph, where you describe consequences of FKF1-GI complex formation on CDF1 and CO degradation, but do not say anything about consequences of ZTL-GI interactions.*

As noted above, we have reworked the Introduction to better focus on the plant circadian and photoperiodic biology. In doing so, the discussion of ZTL-GI interactions has been moved to the Introduction and expanded upon. It is also summarized in Figure 1—figure supplement 2.

*From the Results:*

*Second paragraph:"… a long photocycle would lead to increasing delays in PRR5 and TOC levels leading to progressively longer circadian phenotypes". It is more precise to state "TOC degradation" instead of "TOC levels". Also, the authors should explain or at least provide a reference for the connection between PRR5 and TOC levels and circadian period length.*

The correction has been made and references to TOC1 levels on circadian period length have been added to the Introduction.

*Subsection “LOV dimerization”: "ZTL/FKF1 dimer", this is somewhat ambiguous, as far as we understand that you study ZTL dimers, FKF1 dimers, but no ZTL/FKF1 heterodimers – please clarify.*

We have changed the wording to “ZTL and FKF1 homodimers” to be more specific.

*Discussion of the G46S mutant (Results, fourth paragraph; subsection “V48I disrupts ZTL:GI interactions”, first paragraph; subsection “ZTL variants alter targeted degradation of clock components”, first paragraph; Figure 1—figure supplement 2; Figure 4—figure supplement 1): In the subsection “V48I disrupts ZTL:GI interactions” G46S is reported to "enhance GI complex formation in both the dark and light states". In the same subsection the authors reference Kim WY et al., Nature 2007 for "strong GI-interactions of the G46S mutant". However, Kim et al. describe a non-detectable GI interaction for a ZTL(G46E) variant in Co-IP as well as a slower migration, therefore indicating potentially aberrant folding of full-length ZTL(G46E) in plant material. The authors should discuss the opposite effect of the G46E mutation (Kim et al., 2007) and the G46S mutation (this study) on GI interaction.*

We agree with the reviewers that this is an important observation, but were reluctant to be too speculative. We have included new commentary on the differences between G46S and G46E as well as added an additional supplemental figure (Figure 4—figure supplement 2) to highlight the different structural effects that could be predicted based on G46S and G46E mutations. The new commentary is copied below:

*“*We note that the G46S results reported here diverge from the effects of the G46E mutation reported by Kim et al. (Kim et al., 2007), where G46E abolishes GI interactions due to apparent misfolding of the LOV domain. […] Such results are consistent with enhanced GI interactions in G46S. Combined, the results implicate the G46/V48 locus in ZTL allostery and regulation.”

In regards to Kim et al., 2007, we did not intend on suggesting that Kim et al., discuss G46S, rather we were referencing the difference between light-state and dark-state GI binding. We have removed the reference at that spot to avoid confusion.

*In the subsection “ZTL variants alter targeted degradation of clock components”, the authors state that the G46S mutant is not soluble and could therefore not be purified, crystallized or used for UV/Vis kinetics. Hence, mutation of Gly46 (to S or E) is critical for ZTL folding. Could the slower dark recovery kinetics of the ZTL(G46S/G80R) double mutant (time constant 21 h compared to 1.4 h for wt ZTL and 6.6 h for G80R ZTL) be a consequence of reduced protein stability or aberant folding? The authors should assess protein stability and folding of the ZTL mutants (in particular G46S/G80R) e.g. by thermofluor or CD spectroscopy to validate the kinetics and exclude that potential conformation defects affect GI-binding (Figure 4—figure supplement 1).*

We thank the reviewer for the comments and apologize that our wording led to some confusion. We do predict that the G46S substitution will have a structural effect and act as an allosteric mutant to affect function (see commentary above and Figure 4—figure supplement 2); however, we do not believe this is a stability issue.

Our wording was misleading as it did suggest misfolding/aggregation, but that was not our intention. Rather, the problem is trying to express and purify the protein in *E. coli*. This is a problem with ZTL in general, where working with WT ZTL is nearly intractable and requires 18L of cell pellet or more, for individual experiments. This is exacerbated in the G46S variant, where even with 18L of pellet or more, the protein yield is too poor for biochemical characterization. This is common for most mutations to ZTL, except G80R.

The G80R variant was the first we made. We made it to restore the π-cation interaction that stabilizes the helical surface of PAS proteins (Figure 2). Our idea, and it worked, was that G80R would allow reasonably high protein yields from *E. coli* so ZTL could be studied. The G80R substitution also allows access to variants, through similar stabilization. We do not believe the G46S variant on its own is particularly destabilizing, rather, it makes a nearly intractable situation worse. We have added the following to the Methods section to aid in understanding.

“We note that ZTL is difficult to express and purify from *E. coli* as most WT proteins are confined to inclusion bodies. […] This allows access to G46S:G80R, despite intractable yields of the isolated G46S variant”.

We have also changed the wording regarding G46S being insoluble to now read:

“G46S variants were excluded due to poor yields from *E. coli* that render solution or structural information intractable for the isolated G46S variant (see Methods).”

In regards to CD/thermofluor analysis, we have done thermofluor analysis of WT, G80R and G46S:G80R (see Figure 8). G46S:G80R actually is significantly more thermostable, which may contribute to the longer photocycle, where G46S:G80R demonstrates a higher Tm (66.0 °C) than WT and G80R (both 62.5 °C). Notably, all Tm’s are considerably outside the range of biological relevance.

Author response image 1.**DOI:**
http://dx.doi.org/10.7554/eLife.21646.024

We also note that the temperature (Arrhenius/Eyring) data of ZTL variants also addresses the reviewer’s concerns. We have looked at WT, G80R and G46S:G80R and confirmed that all three demonstrate linear Arrhenius dependency within biologically relevant temperatures (16-40 °C). Further, they all demonstrate the same enthalpy of activation. This is consistent with no gross perturbation of structure/stability throughout biologically relevant temperature ranges.

Having no major effect on structure, but resulting in stabilization of the protein at high temperature is what we would predict from our models of the effect of G46S on ZTL structure (Figure 4—figure supplement 2), where G46 has two effects: 1) Reducing dynamics by addition of a Cβ and 2) Formation of a G46S-Q154 H-bond. Both should stabilize the Ncap, leading to increased stability of this key region.

We agree with the reviewers that this is an important issue to address to verify the LOV domain is not partially unfolded in G46S (like in G46E), particularly given how the initial manuscript was worded in regards to insoluble G46S. However, we do not feel the thermofluor data warrants inclusion in the manuscript and is instead provided here for reviewers and for inclusion in the published response to reviewers. The data itself is better suited to a more comprehensive study of G46S when structures of G46S:G80R and/or FKF1 are available.

*There are distinct migration properties of ZTL mutants (Figure 5) on SDS-PAGE. G46S, G46S:G80R and V48I:G80R migrate differently from the WT, V48I and G80R. Do you have a hypothesis as to the basis of these migration differences?*

We agree with the reviewers that there is a subtle, but distinct difference in the migration properties of those mutants. If forced to speculate about a possible hypothesis, I would lean towards subtle reorientation of the Ncap by the presence of G46S and V48I (see the new Figure 4—figure supplement 2). The G80R may stabilize the helical surface in the double mutants forcing any structural defects to the Ncap.

One may expect V48I alone to have the same migration difference, however it is possible the lack of R80 allows the stress of V48I to propagate to the helical interface. In such a scenario, the absence of the π-cation interaction stabilizing G80R variants, directs conformational changes to the helices instead of Ncap.

Given the highly speculative nature of such explanations, and the subtle differences in migration patterns, we refrain from including commentary on these observations in the published text.

*Can the ZTL-LOV crystal structure explain how the added G80R mutation stabilizes the G80R/G46S mutant compared to the single G46S. It is not intuitive because G80 is on the surface and not close to G46 (and V48). This point is interesting to address, because the correlated G46S, G80R exchange may be a stabilizing co-evolution in the FKF1-LOV domain. Moreover, G80 is only Gly in A. thaliana ZTL, but replaced by R/Q in all other ZTL/FKF1/Phot-LOV proteins aligned in Figure 7.*

We thank the reviewers for their thoughtful comment. The reviewers are correct that it is odd that *A. thaliana* ZTL is unique in containing a Gly residue at this site. Indeed, it is R/Q/K in nearly all LOV proteins. As noted above, we have added brief commentary to highlight the effect of G80R on ZTL proteins and the G46S/G80R double mutant, namely enhanced expression/solubility in *E. coli* (see comments above).

To better address this we have expanded our comments on the initial selection of these sites. We have added the following:

“In contrast, to the best of our knowledge, the G80 position has not been exploited for tuning LOV kinetics. […] Since, G80 is unique to *Arabidopsis* ZTL, and R/Q/K substitutions are permitted in other ZEITLUPE and LOV proteins (see Discussion), it is unlikely to affect allostery and may function as a site to uniquely affect photocycle kinetics.”

We have also added very brief commentary in the beginning of our discussion on ZTL structure:

“The primary difference is the formation of a π-cation interaction that stabilizes the Dα/Eα linkage and C82 that is necessary for C4a adduct formation (Figure 2). Increased rigidity of C82 imposed by the π-cation interaction is consistent with having an effect only on photocycle kinetics.”

The last line was added to help clarify a structural reason, namely, that restoration of the R80 π-cation interaction, stabilizes C82 (C82-flavin C4a adduct) through steric restraints. Similar comments have been added to the legend for Figure 2.

In regards to co-evolution, we do not believe that G80R is involved in coevolution, as most ZTL proteins contain the G80R substitution, but retain G46. We believe this is consistent with the G80 site being specific to kinetics, whereas the G46 position seems to be an allosteric site. As discussed in the text any side chain at position 46 does not allow the exposed conformation of Q154. G46 was evolutionarily selected to allow the unusual Q154 mechanism.

The reasonable question then is, why G80 in *A. thaliana*, if it is destabilizing? The primary effect of G80 appears to be in affecting flavin incorporation, where G80 leads to most ZTL expressed in *E. coli*, to be apo- and relegated to inclusion bodies. G80R leads to better flavin incorporation and makes ZTL tractable. We currently believe that the G80 selection is specific to *A. thaliana* ZTL to optimize stability and photocycle kinetics under typical *A. thaliana* growth conditions. Any loss in stability/FMN incorporation is likely compensated by the chaperone functions of GI. Such an analysis, though, requires detailed kinetics of ZTL/FKF1 proteins in diverse species as well as diurnal expression profiles and in vivostudies. Unfortunately that is beyond the scope of the current study.

*From the Discussion:*

*There are interesting aspects of your work that should be discussed such as:*

*1) Does the crystal structure provide any insights on the co-evolution of G80R and G46S/V48I in FKF1? G80 is Gly only in At-ZTL, this position is R/Q in all other proteins as well as in ZTLs of other species (aligned in Figure 7).*

As discussed above, we do not believe that the G80 site is involved in coevolution or is involved in differentiating ZTL and FKF1 beyond affecting photocycle kinetics. We believe such commentary is best suited earlier in the manuscript. As noted above we have expanded commentary in several locations to address the uniqueness of G80 to *A. thaliana.*

In contrast, we believe G46/V48 are evolutionarily selected to differentiate signal transduction mechanisms present in ZTL and FKF1. Both residues are essential to allowing the heterogeneous orientation of Q154. We have rewritten the discussion to focus on the implications for G46/V48 and the heterogeneous environment of Q154 as we think it is an important aspect of plant biology. We also replaced Figure 7, to show coevolution of G46/V48.

*2) Ncap mutations G46E (Kim et al., 2007) and V48I weaken GI-binding, whereas G46S strengthens GI- binding. This difference, in particular between G46S and G46E should be discussed.*

As noted above, the initial commentary in the Results section has been expanded to highlight this observation. We have also included the new Figure 4—figure supplement 2, to highlight structural perturbations expected from these very different mutations.

In addition, as noted in the preceding comment, we have rewritten the discussion to better highlight the nature of the G46/V48/Q154 site and its role in differentiating ZTL and FKF1 function in plant biology.

*3) The conformation of Q154 in the dark-state structures of wt and G80R are heterogeneous (Figure 3). Does this mean that both the exposed and the buried conformation are present in these structures? Or is it a mixture of different "exposed" conformations, as suggested by the dark state model for wt/G80R ZTL shown in Figure 4 and in the first paragraph of the subsection “ZTL:TOC1/PRR5:GI circuit dictates circadian period through LOV kinetics”? Also, how is the wt/G80R light-state model shown in Figure 4 obtained, since only dark-state structures are available? Some more discussion is needed here. It would be helpful to show a superposition of the wt and G80R dark state Q154 with the V48I/G80R dark- and light-state structures to i) classify the heterogeneous Q154 conformations as exposed or buried and ii) to illustrate the impact of the V48I mutation on the Q154 position. It would also help to describe the heterogeneous G154 conformations of wt/G80R as "exposed/buried" already in the Results section, e.g. the second paragraph of the subsection “N-terminal CGF motif defines a locus of signal transduction that differentiates ZTL from known LOV signaling mechanisms”.*

*Subsection “Insight into LOV allostery and conservation in plants”, first paragraph: Please correlate buried/exposed Gln conformation with light/dark state.*

We apologize for the confusion in both the nature of the heterogeneous orientations and in regards to the source of the model shown in Figure 4.

We also thank the reviewer for the comment and related comment on the oddities in ZTL and FKF1 function, where there seem to be mysteries in regards to the ability of these proteins to perform contradictory tasks (FKF1: CO Stabilization; CDF1 Degradation – ZTL: light-state degradation of CO; dark-state degradation of TOC1/PRR5). They are very keen observations and of central importance.

We have made several edits to address the reviewer’s concerns. First, we have added a Figure 3—figure supplement 1 that contains all four orientations of Q154 for WT and G80R. This figure contains the dark-state V48I:G80R “exposed” conformation and the buried conformation. We believe this highlight the range of conformations observed in the WT/G80R structure.

We have also greatly expanded the discussion on our proposed mechanism of ZTL (see section titled “Insight into ZTL function in *Arabidopsis*”), where the heterogeneous conformations of Q154 are coupled to the evolutionary selection of G46 and V48.

The key is that the selection of G46 and V48 permit the heterogeneous environment of Q154 that essentially dampens light-dark regulation of ZTL, and leads to differences compared to all other known LOV structures. This is consistent with ZTL retaining degradation function under both LL and DD conditions. We propose that dark simply biases ZTL towards a state of enhanced degradation activity.

In V48I, the additional methyl group directs Q154 to the exposed conformation only. No heterogeneity in Q154. Given that dark-state V48I:G80R contains only the exposed conformation and that V48I containing proteins demonstrate high degradation activity, we define this as the dark-state. The buried conformation partially occupied in the V48I:G80R light-state molecule is then consistent with the light-state.

In WT and G80R proteins, G46 (no Cβ) and V48 (no methyl group) lack elements to direct Q154, thus ZTL can sample all conformations, where the population shifts towards the buried-state in the light. This is consistent with LKP2 and *Brassica* ZTL’s, tolerating a L154 that typically abrogates LOV function.

We’ve substantially edited Figure 4 to highlight these mechanisms and to reflect the changes in the discussion. We also then address the following important comment (copied from above):

“Also, how is the wt/G80R light-state model shown in Figure 4 obtained, since only dark-state structures are available? Some more discussion is needed here.”

I had thought about the best way to portray our mechanism for a long time, to allow insight into the mechanism, without being misleading regarding the data source. Typically, I’d avoid using a “sticks” representation of key elements, as it can potentially suggest that it is an actual structure, and the reviewer is correct that we only have dark-state structures of WT/G80R. Based on the reviewer comments, I failed at the first attempt. To avoid confusion, and to address the reviewers concerns and better represent the mechanism summarized above we have redone Figure 4.

Figure 4 now contains three elements. 1) The typical LOV model, derived from light- and dark-state structures of VVD, where the Gln is always buried and only changes H-bond partners. 2) Actual structure derived from WT/G80R showing heterogeneous orientation of Q154. We include either the buried/exposed orientation for reference and use an arrow and text to emphasize selection of bias towards the exposed/buried conformations. 3) The actual structures of V48I:G80R.

We feel this best portrays all aspects of the model. We’ve expanded the legend to include references to all source material.

*Subsection “Insight into LOV allostery and conservation in plants”, second paragraph: "In ZTL, light activation leads to rotation of Q154 to a buried conformation observed in all other LOV structures". "all other LOV structures" implies "all light- and dark state structures of LOV domains that have ever been solved". The authors should be more specific here.*

Indeed, the buried conformation is the orientation of the active site Gln in all light and dark-state structures that have been solved to date. The conformation and mechanism is now also included in Figure 4.

The exposed conformation in ZTL is exceptionally unique highlighting the evolutionary selection of G46/V48. We have rewritten the Discussion entirely to better clarify the uniqueness and role of G46/V48/Q154 in plant biology. The relevant sentence has been edited to better specify that “all known” structures contain the buried conformation.

“In all dark and light-state LOV structures currently in the protein data bank, the active site Gln (Q154 in ZTL) adopts a distinct buried conformation in the dark, with the NH moiety of the Gln side chain near flavin-N5 (Zoltowski et al., 2007, Halavaty and Moffat, 2007)”.

*Your results address the difference in light-regulated target degradation between ZTL and FKF1: FKF1 promotes CDF degradation in the light while ZTL promotes TOC1/PRR5 degradation in the dark. However, in the Discussion it would be helpful to the readers to elaborate on the role of FKF1 in CO stabilization which also occurs in the light as noted in Song et al., PNAS 2014 (and some of the authors of this paper are authors on Song et al., 2014). That FKF1 is involved in stabilization and degradation may be a bit of a mystery, but it is important to note this as a discussion point and, if the structure-function data from this current work lead to any hypotheses as to how light-activated FKF1 stabilizes CO, that would be interesting and valuable for readers.*

We thank the reviewer for their astute observations. As noted above, we have reworked the Introduction and Discussion to better address such mysteries, as we believe they are very relevant to the evolutionary selection of G46/V48 and the heterogeneous Q154.

We now address the following seeming contradictions based on our structures and proposed mechanism in the Discussion:

ZTL/GI destabilizes CO in the light ZTL degrades TOC1/PRR5 in the dark

FKF1 only has light-state functions

As discussed in the edits, we believe that ZTL having light/dark functionality vs. FKF1 only light is a direct result of G46/V48.

We address the question regarding CO stabilization/CDF degradation in the Introduction. The seeming discrepancy here resides in another layer of ZTL group protein regulation, the fact that both the Kelch repeat and LOV domains are involved in protein:protein interactions. It would appear that for FKF1 CO recruitment by the LOV domain, stabilizes CO; whereas CDF is targeted through degradation through the Kelch domain. We envision such competition between domains is likely a large area of regulation and will be a very rich area of study moving forward.

*From Figures/Tables:*

*Figure 1—figure supplement 2: It would be easier for the reader to understand this figure if you annotate the ZTL mutants within the graphs and mention the time constants within the figure legend or to refer to Table 1. Figure 1—figure supplement 2: for base catalyses by imidazole, the authors should refer to Pudasaini and Zoltowski, 2013 to allow the reader to better assess the imidazole effect on ZTL.*

Each panel is now annotated with a label indicating the mutant. The time constants are now added to the legend along with a reference to Table 1.

Pudasaini and Zoltowski, 2013 is now indicated in the legend.

*Figure 2—figure supplement 1: Why was the ZTL (16-165) construct used for MALS instead of ZTL (29-165), which was used for crystallization (Materials and methods) and kinetics (Table 1)? Which constructs were used for analytical gel filtration studies (Figure 2—figure supplement 1, Figure 2, Figure 2—figure supplement 2)?*

We apologize for the lack of clarity in constructs used in SEC experiments. With the exception of Figure 2—figure supplement 1/B, all constructs used in experiments are ZTL 29-165.

For Figure 2—figure supplement 1/B, we do use ZTL 16-165 to allow direct comparison of the equivalent ZTL, FKF1 and LKP2 constructs. All previous characterizations of FKF1 use a 28-174 construct equivalent to ZTL/LKP2 16-165. Due to stability issues involving FKF1 we have to use 28-174 and thus, felt it best to use the equivalent ZTL and LKP2 constructs for the comparison figure.

The apparent MW’s from elution volumes in SEC (now included in the legends for all SEC traces) are large, ~60 kDa, and more consistent with a trimer/tetramer, compared to that of the smaller 29-165 construct (~40 kDa; Figure 2). Given the tetrameric unit cell, we conducted MALS, panel B, to rule out tetramer formation in the longer constructs. MALS confirms a dimer indicating elongation of the ZTL/FKF1 dimer. We have also done MALS on 29-165 to confirm dimerization and the MW from MALS experiments is now referenced in the figure legends.

We want to point out a change to the text though in reference to LKP2 in the figure legend. Over the past 2-3 months we have turned our attention to examining LKP2 structure/function. It has become clear that LKP2 can dimerize in a manner consistent with ZTL/FKF1, but that dimerization is of much lower affinity and highly sensitive to buffer conditions. In this regards, LKP2 may not generate a physiological dimer, to avoid being possibly misleading, we changed the text regarding LKP2 to read:

“LKP2 (16-165 apparent MW=33 kDa) adopts a much smaller hydrodynamic radius indicating distinct differences in oligomeric structure and/or affinity.”

As the manuscript does not discuss LKP2 at all, we considered removing the LKP2 SEC trace, but feel that this distinction may be of interest to the biological community and felt it should remain. The updated wording better emphasizes the current state of the data until a separate comprehensive study is finished.

*Figure 2, Figure 2—figure supplement 1/C, Figure 2—figure supplement 2: For analytical gel filtration studies, please state which SEC columns were used (e.g. S200/S75 10/300, 16/60 etc.) and add elution volumes of standard proteins for reference.*

We apologize that his information was not included. A section regarding the methods for gel filtration studies is now included and contains the requested information.

*Figure 2 legend: MALS in Figure 2—figure supplement 1, not Figure S3.*

Corrected.

*Figure 2—figure supplement 2: why does light-state ZTL-R95A show a lower 280 absorbance than dark-state ZTL-R95A at equal concentration? One possible reason is that light-exposure leads to partial aggregation.*

The lower absorbance at 280 nm is not entirely unexpected. Formation of the light-state adduct in LOV proteins leads to a bleaching of the 280 nm absorption band of the FMN/FAD cofactor. Typically, this is not observed if the data is reported as normalized absorbance at 280 nm (as in Figure 2). Here, we are showing data at two concentrations so we cannot normalize the absorbance and the bleaching of the 280 nm band is evident.

We note that we cannot entirely rule out partial aggregation, however we do not believe any appreciable aggregation is occurring, as no void peaks are observed in the elution profiles following light-treatment.

*Figure 3: Put a gap between 3B and 3D. It is really difficult to distinguish the two panels as it is. Also, please label and show sticks for V48 and G46 in Figure 3 and/or 3B to show their relative location to Q154, FMN and C82.*

A gap was added between 3B and 3D. We have also added sticks and labels for V48 and G46 in Figure 3. We did not add them to Figure 3 as it obscures the alternate conformations of Q154 in that figure.

*Figure 3 legend: with sulfur-pi and pi-pi interactions you probably refer to C45 and F47. It would be helpful for readers to understand your work to explicitly mention "C45" and "F47" as involved residues.*

Reference to C45 and F47 have been indicated as involved residues in the legend for Figure 3 and also in Figure 2/F that involves the same contacts.

*Figure 4: Q154, not Q145.*

The figure has been corrected.

*Figure 4 State clearly in the legend if AsLOV is a dark- or light-state structure? Figure 4: There is no density for FMN. Please explain this.*

We thank the reviewers for highlighting this issue as it turns out the final manuscript figure uses the LOV1 domain of *Arabidopsis* phototropin 1. The legend now correctly indicates this, indicates it is the dark-state and includes the PDB ID.

We also note that FMN density is excluded for clarity. We want to highlight the electron density for the active site side chains that would not be discernable if the FMN density were included. We include a different orientation in panel B that includes FMN electron density.

*Figure 4: the legend not clear as to what is the difference between these two panels. It would be better to describe 4B and 4C separately and be clear on how the two panels are related. Also, FMN in Figure 4 looks a bit distorted. Please comment on this.*

Legends for Figure 4 have been separated. Panel B is included to show electron density for FMN and to show clear electron density for C4a adduct formation. This confirms that we did directly crystalize the light state. This orientation does not allow for clear visualization of the alternative conformations of I48/Q154. Panel C is rotated to allow visualization of the alternative conformations and clear visualization of the Fo-Fc electron density. For this to be clear the electron density for FMN, present in panel B, is excluded. We feel providing both orientations allows the best visualization of all electron density, adduct formation and alternative conformations.

In regards to the distortion of FMN, this is expected. C4a adduct formation induces sp3 hybridization of the C4a carbon and loss of planarity of the isoalloxazine ring. Such distortion is observed in the light- state structures of VVD (Vaidya et al., 2011, Science Signaling; PtAu1a (Heintz and Schlichting, 2016, *eLife*) and PpsB1 (Circolone et al., 2012, J. Mol. Biol.).

*Figure 4: for WT and G80R only dark-state structures are available. On what basis is the model for the WT/G80R light state developed?*

*Figure 4 legend: "Activation of ZTL at dusk.…". It would be more accurate to state that "activation" means dark-recovery of ZTL, i.e. generation of the dark state of ZTL.*

The legend has been edited to reflect the requested changes above:

*“*Predicted divergent ZTL model of allostery and signal transduction based on the integrated structural, mutational and in vivodata. […] For V48I, I48 selects the exposed conformation in the dark and leads to only partial burying of Q154 (shown in C, D), leading to constitutively high ubquitination activity that mimics the dark-state of WT-ZTL.”

We note that we have changed the wording to reflect that Figure 4 WT/G80R light-state is predicted based on integrating the observed structural, mutational and in vivodata. We summarize the logic here and have rewritten the Discussion. As noted in a previous comment, we also have modified the figure considerably, to ensure that the model is clearly portrayed and the source material of any structural information is well defined. We address this above, in regards to a previous comment, but add more commentary here to describe our logic.

We have clear structural evidence indicating an unusual heterogenous orientation of Q154. We also have clear data showing V48I:G80R selects for the unusual exposed conformation of Q154, the site of protein allostery in all characterized LOV proteins. We have solved the crystal structure of the light state using a long-lived mutation similar to what was used to obtain the VVD light-state structure (Vaidya et al., Science Signaling) that enables direct crystallization of the light-state. The light-state molecule shows partial rotation of Q154 that is impeded by the presence of I48. We proposed based on this insight that the light-state adduct drives rotation of Q154 from the exposed to the buried conformation. The latter is present in all solved structures of LOV proteins, now represented in Figure 4. For WT/G80R there is no selection for a distinct conformation, rather we propose activity is based on altering the population of heterogeneous conformations between the buried and exposed conformations. This is consistent with ZTL retaining light/dark functions in vivo, as discussed in the expanded Discussion.

This led to several testable hypotheses. 1) The exposed conformation is essential to dark-state activity and mutations to G46 that allow the conformation would have a functional effect. Indeed G46 leads to enhanced interaction with GI in the dark, consistent with disruption of dark-state signaling. 2) Disruption of rotation of Q154 by I48 would lead to constitutively high dark-state like activity of photoactivated V48I mutants.

This hypothesis is confirmed via both reduced interactions with GI and by enhanced degradation of PRR5 and TOC1 in the light. These effects are physiologically relevant as they alter circadian period. 3) If Q154 is rotating following light exposure, G46 and V48 should have been co- evolutionarily selected for to differentiate ZTL signaling from other LOV proteins. Figure 7 and Figure 7—figure supplement 1 suggest this to be true for both dicots and monocots. Combined, we feel that the updated model of ZTL signaling depicted in Figure 4 is the most consistent mechanism for ZTL function and signaling.

We note that no mechanism can be definitive and have changed the wording to “predicted” to represent that fact. Ideally we would obtain light-state structures of the WT and G80R, unfortunately this is not technically feasible as the lifetimes of the light-state are too short to allow direct crystallization in the absence of continuous light. Exposure to continuous light leads to protein damage and the inability to crystalize ZTL. Such problems have been highlighted in (Vaidya et al., 2011, Science Signaling; and Heintz and Schlichting, 2016, *eLife*) and in some cases can abolish direct light-state crystallization. The alternative is to use mutations to stabilize the light state and use an integrative approach to validate the structural mechanism. Although non-ideal, we feel our data firmly supports our model that indicates a unique-divergent mechanism for ZTL proteins that is evolutionarily selected. Further, we outline new functional roles of LOV chemistry instrumental to plant growth and development.

*Table 2: For easier understanding, please define D = dark, L = light and label WT and G80R structures as "dark". Please also add the number of Ramachandran outliers.*

The table has been corrected as requested. In addition, crystallographic statistics have been updated to reflect the final deposited information.

Some changes in the statistics occurred following removal of some waters.

*Table 2 and Results, end of fourth paragraph: according to Table 2 the V48I:G80R mutant was crystallized in the dark state (2.1 A resolution) and in the light state (2.3 A resolution). In line 172-174 the authors only mention light-state crystals of V48I:G80R and say that the light state structure is at 2.1 A (2.3 A would be correct for the light-state structure). Please correct this discrepancy.*

We apologize for that discrepancy and it has been corrected.

*Table 2: The data for light-state V48I:G80R are only 87% complete. Was there a radiation damage or an adduct cleavage problem during data collection? Please comment.*

In general collecting light-state LOV structures is complicated by radiation damage that leads to cleavage of the photo-adduct (see Zoltowski et al., Science, 2007 and Heintz and Schlichting. *eLife* 2016;5:e11860). The radiation damage can lead to difficulties in properly evaluating a light-state structure. Ideally, multiple data sets are collected at different spots along the same crystal and merged to maintain completeness without damaging the adduct. For the ZTL light state crystal we can only collect data at one spot due to crystal size and morphology. That left the data set of lower than ideal completeness upon truncation to 60-frames, similar as to what is observed for light-state VVD (Zoltowski et al., Science, 2007). To verify that data truncation was not altering our structural conclusions, we did solve the structure with the complete data set (96% complete) using more frames. There are no structural differences between truncating the data set and using the full data set. The only effect is in regards to clearly observing the electron density for the covalent adduct that is reduced upon further radiation exposure. As this is essential to confirming direct crystallization of the light state, the truncated data set (less complete) is used.

*General Comment:*

*Standardize the amino acid nomenclature (1 or 3 letter code?). In most of the text it is the 1 letter code but sometimes the authors use 3 letters; e.g. in the first paragraph of the subsection “Insight into LOV allostery and conservation in plants”.*

We apologize for the errors. Single letter code is now used throughout.

[Editors' note: further revisions were requested prior to acceptance, as described below.]

*[…] Reviewer #3:*

*The authors have adequately responded to my issues and submitted a significantly improved manuscript in content and clarity.*

*I do however regret that my issues regarding the Materials and methods section have apparently not been transmitted to the authors. I would like these parts to be improved before publication, as some method descriptions are too inaccurate or missing. Here, I recapitulate the most important requested changes in Materials and methods, now using line numbers of the revised manuscript:*

*Subsection "Structural analyses":*

*How were the crystals for the light state structure of V48I:G80R-ZTL obtained, i.e. what illumination conditions were used to crystallize the light state?*

The conditions have now been added. The illumination strategy mirrors those published in (Vaidya et al., Science Signal., 2011), where the protein is illuminated to a broad spectrum white light source (150 W) on ice. Saturation of the light-state is then verified using UV-Vis prior to setting crystal trays. The trays are then exposed to light once every 24 hours to maintain population of the light state.

*Subsection "Structural analyses", second paragraph: Cryo is only mentioned for V48I dark and light, not for WT and G80R ZTL.*

We apologize for the exclusion of the other cryo-protectants. They are now included.

Subsection "Structural analyses", last paragraph: The authors say that "WT ZTL was used as search model", presumably for mutant structures. But how was the crystal structure of WT ZTL solved, i.e. which search model was used for MR to get the ZTL WT structure?

The WT structure was solved using the LOV1 domain of *Arabidopsis* phototropin 2. This is now clearly indicated. The remaining structures were solved using the WT structure as a search model.

*Subsection "Structural analyses": The authors should explicitly mention that the crystals contain 4 molecules per asymmetric unit. The authors should also describe the final asymmetric unit content, including which residues are undefined/were not build in the model.*

The Methods were updated to include this information.

*Subsection "Size exclusion chromatography": I appreciate that the authors added a Materials and methods for analytical SEC and MALS in their revised manuscript. Herein, the buffer composition used for analytical SEC and MALS should also be specified.*

The buffer composition has been added. The same buffer was used in all SEC and MALS experiments.

*Subsection "Fitting model to experimental data", third paragraph: "under our lighting conditions" is very imprecise. Please provide a protocol/light source (wavelength/intensity) for ZTL light excitation.*

The lighting conditions have been added.

*Subsection "Model generation":*

*Please provide information about the definition and calculation of the time constant k3 that characterizes ZTL dark recovery kinetics. K3 is not introduced in the subsection "Model generation". Here, k3 is first mentioned in the second paragraph without explanation, then used in equation S8, S10, S12 without definition, and it is not explained how the k3 values at the end of the subsection “Parameter estimation for Figure 1” are defined and how they were obtained.*

We apologize that k_3_ was not well defined. We have updated these sections to specify that k_3_ is the rate constant for adduct decay. We further clarify that all values for k_3_ used in modeling are experimentally derived from the time constants present in Table 1. Where k_3_=1/τ.

*Subsection "UV absorption spectroscopy and kinetics": Please provide a illumination protocol (light source/wavelength/intensity) for ZTL light excitation.*

The protocol has been added.

*Subsection "UV absorption spectroscopy and kinetics": The sentence "Kinetic traces were then fit with a monoexponential decay and kinetic parameters were abstracted." is too imprecise. Please provide a more detailed description of how the decay time constants are calculated from the absorption kinetics and provide the equation used for curve fitting. Which wavelength curve is used for fitting, 450 nm or 478 nm?*

The Methods section has been updated to include this important information.

Briefly, we used both the 450 and 478 nm traces. These traces are fit to a standard exponential decay equation of the form:

where τ=1/k.

The reported time constants are the average of three replicates.